# Latent Concept Disentanglement in Transformer-based Language Models

**Guan Zhe Hong**[*1]    **Bhavya Vasudeva**[* 2]    **Vatsal Sharan**[2]    **Cyrus Rashtchian**[3]
**Prabhakar Raghavan**[3]    **Rina Panigrahy**[3]
[1]University of Oxford    [2]University of Southern California    [3]Google Research
guanzhe.hong@eng.ox.ac.uk, {bvasudev, vsharan}@usc.edu,
{cyroid, pragh, rinap}@google.com

## Abstract

When large language models (LLMs) use in-context learning (ICL) to solve a new task, they must infer latent concepts from demonstration examples. This raises the question of whether and how transformers represent latent structures as part of their computation. Our work experiments with several controlled tasks, studying this question using mechanistic interpretability. First, we show that in transitive reasoning tasks with a latent, discrete concept, the model successfully identifies the latent concept and does step-by-step concept composition. This builds upon prior work that analyzes single-step reasoning. Then, we consider tasks parameterized by a latent numerical concept. We discover low-dimensional subspaces in the model's representation space, where the geometry cleanly reflects the underlying parameterization. Overall, we show that small and large models can indeed disentangle and utilize latent concepts that they learn in-context from a handful of abbreviated demonstrations.

## 1 Introduction

Transformer-based Large Language Models (LLMs) demonstrate remarkable in-context learning (ICL) abilities: with only a handful of source-target demonstrations, they can generalize to new queries without any parameter updates (Brown et al., 2020). These successes hint that the models might be inferring latent rules or concepts implicit in the prompt. Understanding this goes beyond studying a specific question about ICL; it is a key step to deciphering how the attention mechanism encapsulates the influence of prior tokens on posterior tokens, through latent intermediate representations.

Our work takes a systematic view of core questions on how transformers process and use latent concepts to perform ICL. By latent, we refer to unstated variables or rules that are necessary for the computation. We also want to understand how the model represents the implicit functions, going beyond just measuring the accuracy of different tasks. Given the breadth of domains where ICL appears to work, we are faced with a large canvas of experiment design: Is the model inferring an elementary function or latently performing chained reasoning? Are the hidden arguments instance-specific, or abstract and reusable? How is ICL performed when "memorized world knowledge" is entailed, versus when the task relies on more elementary logic?

To explore these diverse dimensions, we design and implement two sets of experiments. The first addresses the question of whether and how transformers resolve hidden, intermediate entities in ICL when "memorized world knowledge" is entailed. An example is mapping an arbitrary city in a country to the capital city of this country. The second studies the structure of representations when the model performs numerical or geometrical computations. For example, the model may need to output the next point in a traversal of a circle or a rectangle. While these tasks are fairly simple, we believe that studying them sheds new light on more fundamental questions, including whether the model is taking shortcuts or performing abstract reasoning in its hidden activations.

---

[*]Equal contribution.

For tasks involving world knowledge, we use pre-trained Gemma-2 models. We apply causal and correlation techniques to study how the hidden activations map to certain key parts of the solution process.

- For the largest 27B model, we find that it relies on *step-by-step composition of latent concept representations* to obtain the result. In particular, we discover that a sparse set of attention heads is responsible for resolving the intermediate latent concept (e.g., the country when going from city to capital, or the company when going from product to headquarter city); the latent concepts exhibit orthogonality in the embedding space. Then, we show that there is a set of heads and MLPs deeper in the model responsible for realizing the concrete output (such as the capital) from the intermediate concept.

- In contrast to the large 27B model, we find that the smaller 2B variant contains a much *weaker and noisier* version of the 27B model's circuit. This corroborates the general wisdom that model size significantly impacts latent-concept disentanglement and composition abilities in the LLMs.

- We also show that, as expected, adding more in-context examples leads to higher accuracy. We attribute this to a strengthening of the model circuit's causal importance and an increase in concept representation disentanglement. In other words, the model more fully utilizes its relevant sub-circuits when we add more ICL examples.

- Additionally, we show that the latent concept representations exhibit causal transferability to naturalistic prompts with open-ended generations, with its strength again scaling with model size.

Following these tasks involving world knowledge, we study quantitative self-contained tasks. This allows us to perform more fine-grained experiments on two-layer models trained from scratch. The ICL tasks here are single-step "arithmetic": add-$k$, Circular-Trajectory, and Rectangular-Trajectory.

- Recent work has identified linear task vectors for problems such as basic arithmetic, single-step reasoning, and linguistic mappings (Todd et al., 2024; Hendel et al., 2023). Our key finding is that models not only have task vectors, but these representations *reflect the geometry of the latent variable*. For example, the task vectors for add-$k$ almost entirely project onto a line. More surprisingly, we can intervene on which function the model computes: interpolating along the task vector line approximately interpolates on the latent parameter $k$ for the task.

- For the Circular-Trajectory and Rectangular-Trajectory tasks, we see a similar geometry in a 2-dimensional space. This aligns with the linear representation hypothesis, but provides more nuanced evidence for it by showing a more continuous parameterization along the representation direction. As with the two-hop task, we provide evidence through both correlational analysis and localization of task vectors in the model's representation.

We designed these settings to be clean enough for controlled experimentation, yet hint at some more general phenomena that deserve further study. For example, models can encapsulate the latent structures of the tasks they learn. Moreover, these structures may be localized and interpretable in the model. We posit that even for much more complex tasks, we will be able to find that the latent concepts are captured by a sparse set of attention heads or a relatively low-dimensional encoding.

## 1.1 PRELIMINARIES

To focus on analyzing whether and how transformers utilize latent concept representations for solving ICL, we work with prompts that contain *demonstration-only* specifications of latent functions. Intuitively, we have a source $x$ and target $y$, along with a hidden function $F$. As input, the model only sees a handful of pairs $(x_i, y_i)_{i=1}^n$ where $y_i = F(x_i)$. Then, on a new $x'$ the model has to compute $F(x')$ to obtain a correct answer. The subtle part is that $F = R \circ C$, where $C$ maps input $x$ to a low-dimensional "concept space", which is then refined by $R$ to produce the output.

The concept map is marked by its re-usability over different instances of sources and targets: for example, a function that maps cities and landmarks to Country representations, or one that maps points on a circle to the same radius representation. Such *abstract* representations enable lower-complexity inference. To test for the existence and utilization of such maps in transformers performing ICL, we consider several varieties of demonstration-only prompts, under two categories. The first category focuses on concept maps relying on world knowledge, with Countries and Companies serving as

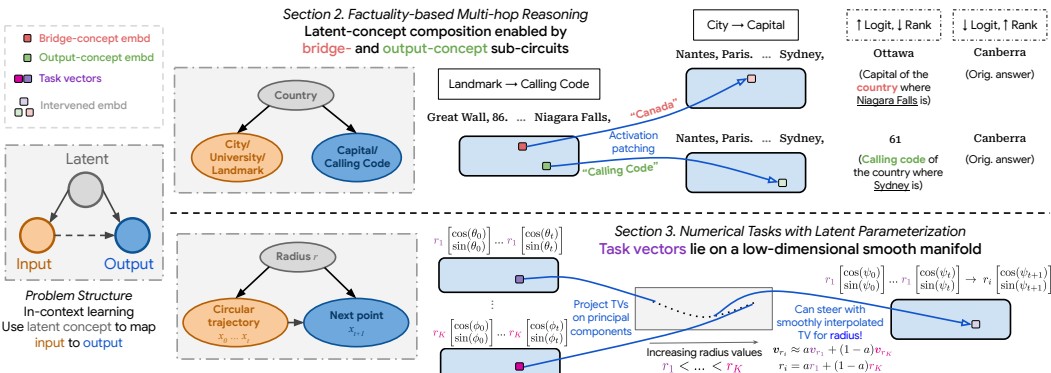

Figure 1: An illustration of our main findings. We primarily focus on how decoder-only transformer-based language models disentangle and manipulate latent concepts for solving in-context learning (ICL) problems. In the **discrete multi-hop** setting, we discover that transformers *compose* latent concept representations for predicting the answer. For example, as shown in the upper half of the figure, by intervening on certain "bridge-concept" attention heads, we can push the model's "belief" (reflected in logit and rank) in the *original answer* Canberra (the capital of Australia, which the city Sydney belongs to) to the "type-corrected" *alternative answer* Ottawa (the capital of Canada, which the landmark Niagara Falls belongs to). In the **continuous-parameterization ICL** setting, we discover that transformers' hidden embeddings capture the *geometry* of the latent concepts for our prediction tasks. For instance, for a transformer trained to predict circular trajectories whose radius is randomly chosen from a continuous interval, not only do we obtain causal evidence for *task vectors* (TVs) which control the trajectory's *radius*, but they also fall on a *smooth* 2D manifold.

the hidden concepts connecting the source to the target (Fig. 1 top half, § 2). The second category focuses purely on self-contained numerical puzzles, with Radius, Offset and other numerical values serving as the hidden intermediate concepts (Fig. 1 bottom half, § 3).

More precisely, we consider the following ICL settings:

- **Factuality-based 2-hop Reasoning.** This setting involves demonstration examples with *under-specified reasoning steps and discrete latent concepts*. Specifically, we consider 2-hop factual recall tasks where the model must map a "source" entity to a "target" entity (e.g., (non-capital) city→capital) by first latently inferring the hidden "bridge" entity (e.g., country), allowing us to ask whether the LLM first resolves the "bridge", then refines it to a "target" entity via (causal) concept compositions. An instance looks like "`Toronto, Ottawa. Mumbai, New Delhi. Shanghai, `", where the answer is `Beijing`. The motivation here is to understand what the model does *after* reading the input city: does it jump directly to the capital, or does it first invoke the country as an intermediate reasoning step? While both are plausible strategies, only the latter captures the latent causal structure that goes through the latent variable (the country).

- **Numerical Tasks with Latent Parameters.** We generate demonstrations based on quantitative parameters. We will see that this determines task similarity and induces smooth geometric relationships in the task space, suggesting that the model encodes such parameters along a low-dimensional geometry. We first consider the numerical *add-k* task (also studied by (Hu et al., 2025)) where the demonstration examples are $(x_i, y_i)_{i=1}^n$ with $y_i = x_i + k$, where $k$ is the latent parameter. An example of add-4 is "`5, 9. 3, 7. 1, 5. 2, `", where the answer is `6`. We then study geometric tasks, such as Circular-Trajectory where points lie on a circle of varying radii and a related Rectangular-Trajectory task. Our goal is to localize, and more importantly, understand the geometry of the task vectors: does it reflect the geometry of the latent parameter?

Both sets of tasks are *demanding enough* to require intricate latent concept disentanglement and manipulation, yet *sufficiently controlled* to permit causal, feature and circuit-level analysis.

## 1.2 Related Work

**In-context Learning Interpretation.** ICL abilities of transformer-based models were first observed by Brown et al. (2020), which sparked work in analyzing this ability. This includes analyzing how pretrained LLMs solve ICL tasks requiring abilities such as copying, single-step reasoning, basic linguistics (Olsson et al., 2022; Min et al., 2022; Zhou et al., 2023; Hendel et al., 2023; Todd et al., 2024; Yin & Steinhardt, 2025a), and smaller models trained on synthetic tasks like regression (Garg et al., 2022; Von Oswald et al., 2023; Akyürek et al., 2023; Bai et al., 2023; Guo et al., 2025), discrete tasks (Bhattamishra et al., 2024), and mixture of Markov chains (Edelman et al., 2024; Rajaraman et al., 2024; Park et al., 2025a; Deora et al., 2025). These setups enable discovery of relations between in-context and in-weight learning (Lin & Lee, 2024; Singh et al., 2025; Russin et al., 2025), and internal algorithms that models implement (Olsson et al., 2022; Edelman et al., 2024; Li et al., 2023; Park et al., 2025a; Yin & Steinhardt, 2025a). We contribute to this line of work, by shedding light on how transformers solve ICL problems which have more intricate latent structures.

**Linear Representation Hypothesis (LRH).** Our results are also connected to the LRH, which essentially speculates that LLMs represent high-level concepts in (almost) linear latent directions (Park et al., 2024; Merullo et al., 2024; Park et al., 2025b; Huh et al., 2024; Ilharco et al., 2023; Li et al., 2025a; Dumas et al., 2025; Beaglehole et al., 2025). Many papers motivated by the LRH then find "concept" vectors that can capture directions of truthfulness (Marks & Tegmark, 2024; Arditi et al., 2024), sentiment (Tigges et al., 2024), humor (von Rütte et al., 2024), toxicity (Turner et al., 2025), etc. We deepen this study, asking how LLMs' representations capture/disentangle latent concepts, and compose them during inference. In addition, the LRH is rooted in the field of mechanistic interpretability, which aims to reverse engineer mechanisms in transformer-based LMs (Elhage et al., 2021; Olsson et al., 2022; Singh et al., 2025; Wu et al., 2023; Wang et al., 2023; Hong et al., 2024; Brinkmann et al., 2024; Bakalova et al., 2025; Heindrich et al., 2025; AlKhamissi et al., 2025b; Vig et al., 2020; Baek & Tegmark, 2025; Wang & Xu, 2025).

**Task and Function Vectors.** A specific line of work in analyzing ICL mechanisms focus on task or function vectors. They show that there exist certain causal patterns which capture the input-output relationship of the ICL task, on relatively simple problems such as "Country to Capital", "Antonyms", "Capitalize a Word" (Todd et al., 2024; Hendel et al., 2023; Davidson et al., 2025; Yin & Steinhardt, 2025b). Similarly, Liu et al. (2024); Merullo et al. (2024); Li et al. (2025b); AlKhamissi et al. (2025a) observed that LLMs compress certain task or context information into sparse sets of vectors. We work with ICL problems with more complex latent structures, and our focus is not solely on (high-level) task vectors, but on how the model disentangle and manipulate latent concepts useful to answering the query. In addition, our work complements the function vector analysis from contemporaneous work of Hu et al. (2025), providing add-$k$ results for smaller models where we have full control over training and can hence conclude that the geometry of the task vector only arises from the latent task structure. We also compare results from add-$k$ with other ICL tasks, giving additional insights.

## 2 Disentanglement of Latent Concepts in 2-hop Reasoning

We start with a 2-hop task based on connecting two facts through a latent entity, where the model must infer the relationships from in-context demonstrations. This task builds on prior work that analyzes problem with a single step of reasoning (1-hop) over world knowledge, such as geography puzzles "Country→Capital" or "National Park→Country" (Minegishi et al., 2025; Todd et al., 2024; Yin & Steinhardt, 2025a). Extending to two steps enables us to understand how LLMs solve ICL problems which have *hidden reasoning steps*, and understand whether they decompose the source-to-target function $F$ into latent maps $C$ and $R$ (introduced in §1.1) using certain internal components.

Our goal is to mechanistically analyze how a pre-trained LLM solves a "source→target" problem when the "bridge" concept is hidden. The model needs to understand the nature of the bridge from the demonstrations, which only contain the sources and targets. Interestingly, models achieve reasonably high accuracy with sufficient in-context demonstrations. This leads to two competing hypotheses:

- *Hypothesis 1 (Shortcut):* The LLM maps the queried input directly to the answer (going from source to target), without internally computing the bridge entity in a discernible manner.

- *Hypothesis 2 (Latent two-hop):* The LLM first resolves the latent bridge concept (e.g., "country") in its hidden representations, then composes it with the output concept to obtain the answer, effectively computing the source-to-target function $F$ as $R \circ C$.

We first present the problem definition and experimental set-up. Then, we present our main causal and correlation evidence favoring Hypothesis 2.

**The "Source→Target" Problem.** We create two-hop ICL puzzles by composing two facts linked by a common "bridge" entity. That is, we sample fact tuples $\{(S_i, r_1, B_i, r_2, T_i)\}_{i=1}^n$, where the *source* entity $S_i$ is related to the bridge entity $B_i$ via relation $r_1$, and $B_i$ is related to the *target* entity $T_i$ via $r_2$. We then create the ICL puzzle in the form $[S_1, T_1. S_2, T_2 \ldots S_n,]$ [Answer: $T_n$][1]. Note that the bridge entities $B_i$'s are *never* specified in the prompt. An example City→ Capital problem is

> `Sydney, Canberra. Nantes, Paris. Oshawa,`

Here, $r_1$ is "belongs to the country of", the (unspecified) bridge entities for this example are "Australia", "France", "Canada", and $r_2$ is "has capital". Therefore, the prompt's answer is `Ottawa`, the capital of Canada, the country `Oshawa` is in.

In the main text, we work with geography puzzles, with source types {City, University, Landmark}, and output types {Capital, Calling code}. In addition to this Country dataset, we work with the Company dataset in App. A.5 for generality[2]. See App. A.2 for details on the data generation process.

We focus on Gemma-2-27B (Gemma Team, 2024) in the main text. To establish a baseline, we evaluate Gemma-2-27B on Source→Target ICL puzzles. Fig. 2 reports its accuracy: although the model finds these tasks more challenging than one-hop counterparts—reflected by the need for more in-context examples—it achieves high accuracy at 20 shots.

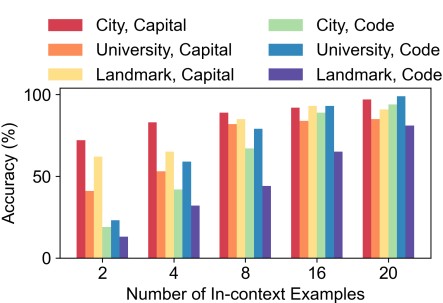

Figure 2: Accuracy of Gemma-2-27B on the two-hop "Source→Target" ICL problems.

We provide new causal and correlational evidence suggesting that the model indeed performs latent multi-hop reasoning via sequential *concept composition*: it first infers an abstract bridge concept representation (e.g. a "Canada" concept), and then specializes to a specific output type (e.g. the "capital" of "Canada") deeper in the model. Surprisingly, we find that the bridge-resolving mechanism is enabled by a sparse set of attention heads in our set of problem settings.

**Methodology.** We use causal mediation analysis (CMA), also known as activation patching (Pearl, 2022; Zhang & Nanda, 2024) to obtain causal evidence for our claims (see App. A.1 for details). We discuss the bridge-resolving mechanism in the main text, and delay the analysis of the output-concept component to App. A.2. For concreteness, consider two prompts with different source-target types:

- A normal prompt, with type [City→Capital]: "Okinawa, Tokyo. Sydney, Canberra. Chicago, " [Answer: Washington; Bridge: USA].
- An alternative prompt, with type [Landmark→Calling Code]: "Chapel bridge, 41. The Grand Canyon, 1. The Great Wall, " [Answer: 86; Bridge: China].

We perform activation patching on normal and alternative problem pairs with different bridge entities, across source-target types, at the final token position. Our experimental hypothesis is that if a (set of) model component effectively serves as the hidden concept map $C$, that is, it computes an abstract representation of the bridge, then this representation should *transfer across different source and target types*. When we run the model on the normal prompt, and replace a selected component's activation by the corresponding activation obtained on the alternative prompt, the model should favor the alternative prompt's bridge *without* bending the output type. For our example, the alternative-to-normal "patching" pushes the model's answer on the normal prompt from Washington to Beijing

---

[1]We think of this as a systematic ICL version of TwoHopFact (Yang et al., 2024). Here, the model must figure out relations between the *source-bridge* and *bridge-target* facts from the ICL examples.

[2]For our Country & Company data, we collect source and target entity data for 40 countries, 36 companies.

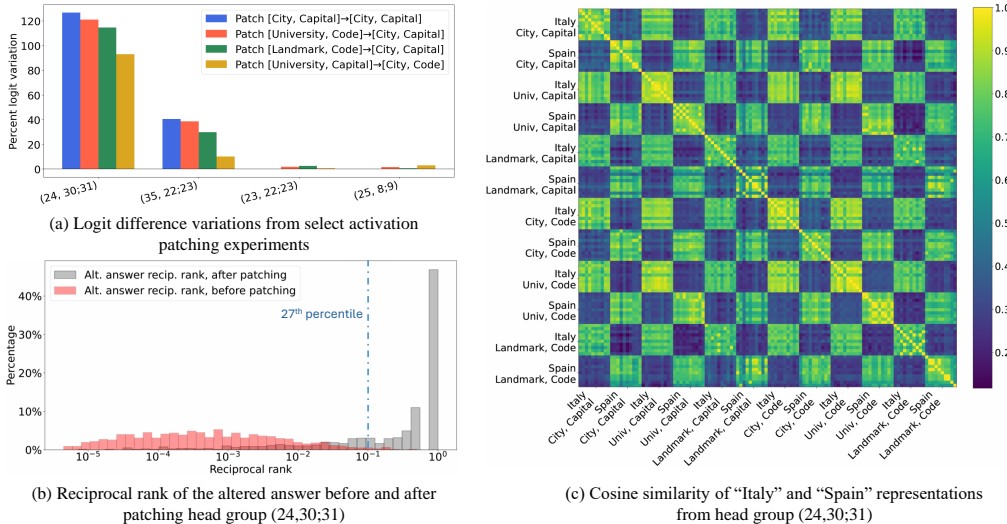

(a) Logit difference variations from select activation patching experiments

(b) Reciprocal rank of the altered answer before and after patching head group (24,30;31)

(c) Cosine similarity of "Italy" and "Spain" representations from head group (24,30;31)

Figure 3: We present causal evidence for the bridge-resolving mechanism in (a) and (b), and correlational evidence in (c) for Gemma-2-27B. In (a) and (b), we run activation patching experiments across prompts with problems of different source and target types, at the final token position. (a) shows the percentage logit difference variation of attention heads with the strongest causal influence, on select intervention experiments. Head group (24,30;31)'s representation exhibits strong transferability across source and target types. (b) further examines the causal role of (24,30;31)[3], in the patching experiment [University, Code]→[City, Capital]. Note that the reciprocal rank of $10^{-1}$ for the type-corrected alternative answer after patching (24,30;31) is in the 27th percentile ($> 73\%$ in top 10 after intervention). In (c), we show a cosine similarity plot of bridge-representation disentanglement for head group (24,30;31), computed on a collection of samples with several combinations of source and target types, and with "Italy" and "Spain" as the bridge values in the prompts.

(the output type remains Capital, but the bridge switches from USA to China). We refer to this as the "type-corrected" version of the alternative answer. This evaluation is slightly unorthodox: rather than judging an intervention by whether it reproduces the literal alternative answer (86), we assess movement toward the type-corrected alternative (Beijing). By contrast, under the *Shortcut hypothesis*, patching activations would at worst break the model and yield nonsensical outputs, or at best, push it to emit the literal alternative answer (86).

We now formalize the intervention with CMA, which enables us to localize components inside the LLM that specialize in handling the different (latent) reasoning steps of the problem.

***Causal Mediation Analysis (CMA).*** Formally, let the normal and alternative prompts be denoted as $\boldsymbol{p}_{\text{norm}} = [S_1^{(\text{norm})}, T_1^{(\text{norm})} ... S_n^{(\text{norm})},]$ and $\boldsymbol{p}_{\text{alt}} = [S_1^{(\text{alt})}, T_1^{(\text{alt})} ... S_n^{(\text{alt})},]$. Let $\hat{T}_n^{(\text{alt})} = \text{Type}_{\text{norm}}(T_n^{(\text{alt})})$ denote the type-corrected output. We run the LLM on both the prompts and cache the LLM's hidden activations at the last token position, denoted as $(\boldsymbol{h}_{\text{norm}}, \boldsymbol{h}_{\text{alt}})$. To perform CMA on a model component of interest, say an attention head with activations indexed by $(\boldsymbol{a}_{\ell,h}^{(\text{norm})}, \boldsymbol{a}_{\ell,h}^{(\text{alt})})$, we run the LLM on the normal prompt again, but this time, intervene by replacing its normal activation with the alternative $\boldsymbol{a}_{\ell,h}^{(\text{alt})}$, and let the remainder of the forward pass execute. We then contrast the model's output behavior on the normal and patched runs by, for example, examining the logit differences between the normal and (type-corrected) alternative answer, and their ranking pre- and post-intervention. Further details on CMA and its suitability to our problem setting are deferred to App. A.1.

## 2.1 RESULTS AND ANALYSIS FOR THE BRIDGE-RESOLVING MECHANISM

***Causal evidence.*** Fig. 3 provides evidence favoring the Latent Two-hop Hypothesis rather than the Shortcut Hypothesis: we observe causal transferability of the bridge representation. In Fig. 3(a),

---

[3]We denote attention head groups with (layer index, head index 1; head index 2; ... head index k).

we report results on a select set of patching experiments: on *both* tasks with and without overlap in source and target types, we observe that a sparse set of attention heads consistently exhibits very strong *causal* effects in pushing the model's "belief" from $T_n^{(\text{norm})}$ towards $\hat{T}_n^{(\text{alt})}$ (reflected in logit difference); the head group (24,30;31) is especially dominant.[4] To further understand whether intervening on (24,30;31) during the normal run really boosts the type-corrected alternative answer $\hat{T}_n^{(\text{alt})}$ (instead of only decreasing model's confidence on the normal answer $T_n^{(\text{norm})}$, which logit difference might not tell), in Fig. 3(b), we show an example patching experiment result of [University, Code]→[City, Capital]. Surprisingly, at least 73% of the time, patching this head group boosts the model's rank of the alternative prompt's answer into top 10 (and directly become the top-1 answer more than 40% of the time!), when its original, intervention-free rank is typically in the hundreds to thousands. In fact, we observe a highly similar trend across all combinations of source and target types on our dataset (delayed to App. A.2).

*Correlational evidence.* To understand the nature of (24,30;31)'s output embeddings better, in Fig. 3(c), we visualize an example cosine similarity matrix of this attention head, with either "Italy" or "Spain" as the bridge values for $S_n$ (the query source entity) in the prompts, across a total of 12 different combinations of bridge, source and target types. Specifically, for each combination of the bridge and source-target type shown in the grid, we sample 10 prompts which obey such requirement,[5] giving us a total of 120 prompts. We then obtain head group (24,30;31)'s embedding of these prompts at the last token position, and compute the pairwise cosine similarities. Observe that the embedding consistently exhibits *strong disentanglement* with respect to the bridge value in the prompt, *regardless of source and target types*. We provide detailed statistics of disentanglement strength in App. A.3 and Fig. 27, where we discuss its relation with the number of ICL examples.

The causal and correlational evidence suggest that there is a small set of components in the model which effectively serve as the latent concept map $C$: their activations on the queried entity are causally *transferable* across source and target types, and exhibit strong intra-bridge-concept clustering and inter-bridge-concept orthogonality (low dimensionality of the representations). To further corroborate the observations made in the main text, we present detailed activation patching results which sweep all combinations of source and target types in Appendix A, spanning Figures 12 - 21 and 27.

**Additional Insights.** We highlight some additional mechanistic insights below (see Appendix for details). First, *the Gemma-2-2B model has a weak and noisy version of the 27B's bridge resolving mechanism*. This suggests that increasing model size likely *benefits* latent concept disentanglement and utilization in the LLM. We provide details in App. A.2, and Fig. 26. Second, in App. A.3, we show that disentanglement strengthens with more ICL examples, which is reflected in both the causal importance of the bridge-resolving components and the angular (dis-)similarities of the bridge embeddings. Lastly, to complement our study on the geography dataset, we experiment with a similar, albeit smaller-scaled Company dataset in App. A.5, where company name is the bridge entity. We again find causal and correlational evidence for the presence of a bridge-resolving mechanism in the model, and observe *overlap* in the attention heads driving this mechanism on the two datasets. Furthermore, to probe whether our findings extend beyond the synthetic setup, Appendix A.4 presents a small-scale study where the same concept embeddings are used for intervening the model in naturalistic prompts. Injecting these vectors tends to coherently steers open-ended generations toward the target country or entity type while preserving fluency, providing preliminary evidence that the learned concepts are not merely puzzle-specific.

## 3 DISENTANGLEMENT FOR NUMERICAL LATENT VARIABLES

In this section, we consider two problems with numerical or continuous parameterization. For these experiments we study a very small transformer, with a similar architecture to GPT-2 (Radford et al., 2019). We use a 2-layer 1-head transformer, with embedding dimension 128, trained with AdamW.

*add-k Problem.* Each task is a sequence consisting of pairwise examples $\{(x_i, y_i)\}_{i=1}^{n+1}$, where $y_i = x_i + k$, for a given offset $k$. Here, we use integer inputs and offsets; all values are in $\{0, \ldots, V-1\}$, each treated as a distinct token. We consider a collection of $K$ tasks parameterized

---

[4] In our CMA experiments, we account for grouped-query attention by patching heads in groups of 2 on Gemma-2-27B. We noticed that this tends to produce stronger causal effects than with individual heads.

[5] We only specify the bridge entity for $S_n$ in the prompt; the bridge for $S_i$ for all $i < n$ are randomly chosen.

by different offset values in $\{k_i\}_{i=1}^K$, where $k_1 = 1$ and we fix $k_{i+1} - k_i = 3$. The model is trained autoregressively to predict the label for each example in the sequence. At test time, the model observes the first $n$ examples and should predict $y_{n+1} = x_{n+1} + k$ for the last example. Here, the latent concept map $C$ needs to produce the offset representation.

***Circular-Trajectory Problem.*** Here a task consists of a sequence $\{\mathbf{x}_i\}_{i=1}^{n+1}$ of points on a circle centered at the origin. Each task is parameterized by the circle's radius $r$; for $K$ tasks, the set of radii $\{r_i\}_{i=1}^K$ is sampled uniformly from $[1, 4]$. A task sequence is generated as follows. We first sample $\theta_0$ uniformly at random in $[0, \frac{\pi}{2}]$, so $\mathbf{x}_1 = r[\cos\theta_0, \sin\theta_0]^T$. Then, we select the *period* $p$ randomly from $\{2, 3, 4\}$, which determines the number of equal consecutive step-sizes. Specifically, we first sample a sequence of $\lfloor\frac{n}{p}\rfloor + 1$ unique step-sizes uniformly between $[0, 1]$, and then get the full sequence of steps $\{a_i\}_{i=1}^n$, where $a_j = a_{j+1} = \cdots = a_{j+p-1}$ for $j \in \{0, p, 2p, \dots\}$. Here, context length $n = 12m + 1$ for integer $m$. We also sample $c \in \{\pm 1\}$, which denotes if the trajectory is clockwise or counterclockwise. Next, we generate a sequence of angles $\{\theta_i\}_{i=1}^n$, where $\theta_i = \theta_0 + c\frac{2\pi}{n}\sum_{j\le i} a_j$. Using the sequence of angles, we generate the sequence, $\mathbf{x}_{i+1} = rR(\theta_i)\mathbf{x}_i$, where $R(\theta)$ is the 2D rotation matrix for $\theta$. Fig. 4 shows an example. As in the previous problem, we train the transformer autoregressively on these types

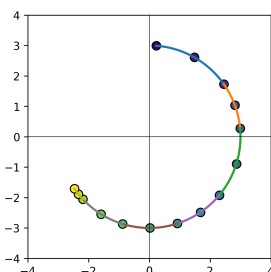

Figure 4: Illustration of an input sequence for the circle trajectory problem. Here, radius $r = 3$, period $p = 2$, sequence length $n = 13$. Every $p$ consecutive steps on the trajectory are equal. We first sample $\lfloor\frac{n}{p}\rfloor + 1$ unique step-sizes in $[0, 1]$, and get the full sequence $\{a_1, a_2, a_3, a_4, \dots\}$, where same colors denote equal step-sizes. Then, we generate the trajectory by rotating point $\mathbf{x}_i$ clockwise by angle $a_i \cdot \frac{2\pi}{n}$ (see text for formal description).

of sequences. Additionally, in App. B, we consider another shape problem, namely the Rectangular-Trajectory problem, where the trajectories contain points on lying on axis-aligned rectangles centered at the origin. The Circular-Trajectory problem is parameterized by one continuous parameter (radius), whereas the Rectangular-Trajectory problem has two parameters, namely the lengths of the two sides of the rectangle. Here, the latent concept map $C$ needs to produce the Radius or Length-Height representations.

## 3.1 Results

***Existence of Task Vectors.*** We first outline the process to identify the task vectors for the add-$k$ problem. We set $V = 100$, $n = 4$, and $K = 2$. Fig. 5 shows the cosine similarities between the layer-2 attention embeddings at the last position for 200 input sequences from each of the two tasks. We observe clustering between intra-task embeddings. Hence, the model disentangles the concept of different offset values in its representation. To provide more evidence for disentanglement, and to locate where the task vectors emerge in the model, we linear probe the embeddings of the model to predict the final output and the offset/task type. We probe embeddings from the output of the MLP at the first layer, the attention block at the second layer, and the hidden and output layers of the MLP at the second layer. The results are in Fig. 6. We see that task type is disentangled at layer-2 attention, and the output is computed at layer-2 MLP. For each task, we treat the layer-2 attention embeddings averaged across 200 input sequences from that task as the task vector.

We visualize the task vectors by performing PCA and projecting them onto the first two principal components; Fig. 7 presents the task vectors for the add-$k$ problem, for $K = 4, 8, 16$ tasks/offsets. In all three settings, the vectors lie on a 1D linear manifold. More than $99.9\%$ of the variance is explained by the first PC. Notably, the model compresses the concept of offsets into a line with the ordering of the offsets (lower to higher) preserved on the manifold (left to right). To corroborate these results with causal evidence, we study the effect of **steering** using the task vectors. Specifically, let $\mathbf{t}_1$ and $\mathbf{t}_K$ denote the task vectors for offsets $k_1$ and $k_K$, respectively. Then, for offset $k_1$ ($k_K$), we steer with $(1 - \beta)\mathbf{t}_1 + \beta\mathbf{t}_K$ ($(1 - \beta)\mathbf{t}_K + \beta\mathbf{t}_1$) and evaluate the accuracy for predicting the output based on the original offset $k_1$ ($k_K$), the 'opposite' offset $k_K$ ($k_1$), or the target offset $(1 - \beta)k_1 + \beta k_K$ ($(1 - \beta)k_K + \beta k_1$), where $\beta \in [0, 1]$. We use the first/last offsets to interpolate them and steer towards intermediate offsets. Fig. 8 presents the top-1 and top-3 accuracies for each case. High

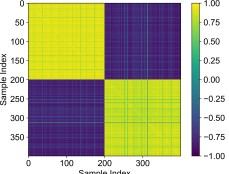 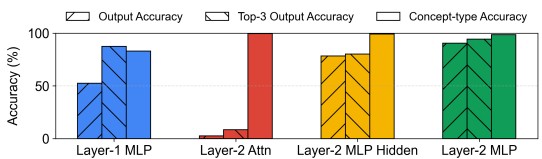

Figure 5: Cosine similarities between the layer-2 attention embeddings for 200 input sequences from two tasks/offsets for the *add-k* problem. Strong clustering between intra-task embeddings shows that the model disentangles the concept of different offsets.

Figure 6: Results for linear probing the embeddings of the trained model at various locations to predict the final output and the task type for the *add-k* problem. The task type becomes disentangled at layer-2 attention, and the output is computed in layer-2 MLP.

top-1 accuracies and $\approx 100\%$ top-3 accuracies for the target for all considered values of $\beta$ indicate that the model output is steered toward the target. Interpolating along the top principal direction is successful at interpolating values of $k$ in the task space, showing that the model captures the concept's geometry. In other words, the layer-2 attention head is serving as the latent concept map $C$ in this setting, mapping input tuples to the offset parameter.

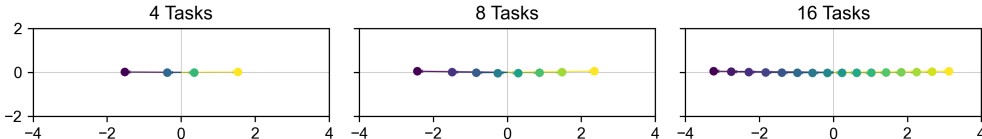

Figure 7: 2D PCA projection of the task vectors for the *add-k* problem. The task vectors lie on a 1D linear manifold. Here the number of tasks refers to the number of values of $k$.

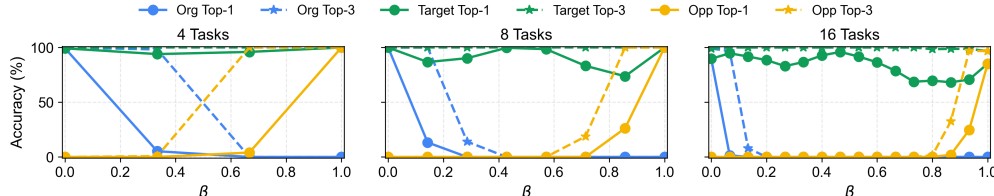

Figure 8: Steering with the task vectors for tasks $k_1$ and $k_K$ for the *add-k* problem (see text for details). We plot the top-1 and top-3 accuracies for predicting the output based on the original offset $k_1$ ($k_K$), the 'opposite' offset $k_K$ ($k_1$), or the target offset $(1 - \beta)k_1 + \beta k_K$ ($(1 - \beta)k_K + \beta k_1$), where $\beta \in [0, 1]$, The result shows that the model output can be steered toward the target.

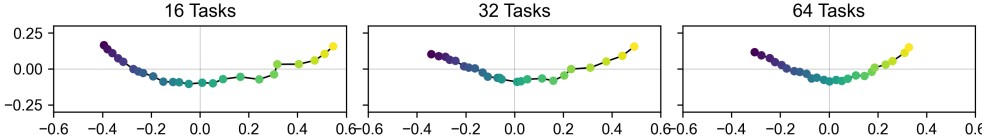

Figure 9: 2D PCA projection of the task vectors for the Circular-Trajectory problem. The task vectors lie on a smooth low-dimensional manifold. Here the number of tasks refers to the number of radius values used for training.

Fig. 9 presents the 2D PCA projection of the task vectors for the Circular-Trajectory problem, for $K = 16, 32, 64$ (training) tasks/radii. In this setting, we consider $K = 24$ radii, spaced evenly between $[1, 4]$ to visualize the task vectors, since this task is continuous. We observe that in all three

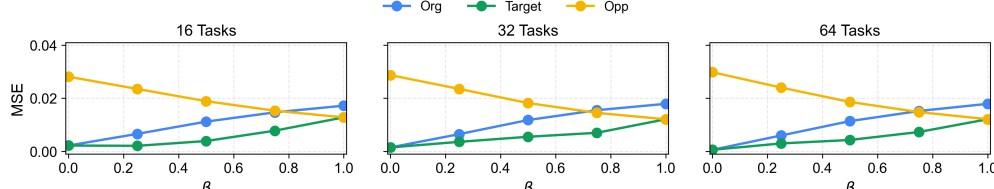

Figure 10: Steering with the task vectors for tasks $r_1$ and $r_K$ for the Circular-Trajectory problem (see text for details). The MSE between the radius inferred from the model output and the original radius $r_1$ ($r_K$), the 'opposite' radius $r_K$ ($r_1$), or the target radius $(1 - \beta)r_1 + \beta r_K$ ($(1 - \beta)r_K + \beta r_1$), where $\beta \in [0, 1]$, indicates that the model output can be steered toward the target.

settings, the task vectors lie on a low-dimensional manifold. The variance explained by the first two PCs in the three cases is $97.05\%, 96.44\%, 93.68\%$, respectively. Similar to the previous setting, the order of the radii (lower to higher) is preserved in the compressed representation.

Fig. 10 presents the results for steering the model output using the task vectors for radii $r_1$ and $r_K$. We follow the same procedure as in the add-$k$ problem, with a different evaluation metric. We compute the norm of the generated output after steering as the model's radius (since the center of the circles is fixed at the origin), and consider the MSE between these radii and the original radius $r_1$ ($r_K$), the 'opposite' radius $r_K$ ($r_1$), or the target radius $(1 - \beta)r_1 + \beta r_K$ ($(1 - \beta)r_K + \beta r_1$), where $\beta \in [0, 1]$, averaged over 200 sequences from each task. We observe that the MSE with the target radius is the lowest, which indicated the task vector can steer the model's output toward the target.

In App. B, we examine the task vectors for the Rectangular-Trajectory problem. Here the model has to reason over two latent continuous parameters, which are the real-valued side lengths of the unknown rectangle. We find that the first 2 PCs lie on a two-dimensional manifold in that case as well. This provides a second example of how the model captures the underlying geometry in terms of both separating the two orthogonal parameters and smoothly interpolating across hidden trajectory shapes.

## 4 DISCUSSION AND CONCLUSION

Our work provides causal and correlational evidence that transformer-based models *disentangle* and *manipulate* latent concepts from in-context demonstrations in a structured, interpretable manner. For two-hop tasks, models contain sparse sets of attention heads responsible for first inferring the bridge entity and then resolving the output. For numerical tasks, the model uses task vectors which closely capture the underlying parameterization. Overall, we hope that results from our controlled experiments can serve as a stepping stone for future empirical and theoretical analysis of transformer models; we discuss potential theoretical implications of our results in Appendix C.

**Limitations and future directions.** There are a few limitations of and possible directions to extend this work, which we discuss below:

- The latent concepts that we test for all have easy to understand human representations, such as countries, companies, numbers, or geometric shapes. There may be other, more intricate, relationships between the ICL examples that the model is also representing in some way. It would be ideal to provide ablations over the human-interpretability of the provided examples, to see if this affects how the model represents the latent concepts.

- The ICL problems have unique answers instead of allowing for ambiguity (e.g. many-to-many instead of many-to-one mappings, or maps involving multiple possible bridge concepts). While this setup enables us to conduct systematic experiments and obtain clean mechanistic insights, it might limit our understanding of how certain concepts could occur in "superposition" in the LLM's representations in some scenarios.

- Future work could leverage the identified concept representations directly for steering model behavior to improve ICL performance or by composing known concept vectors to enable zero-shot generalization to novel compositional tasks. The identified concept embeddings can also serve as probes to inspect which latent concepts are used during reasoning; this approach can complement text-level monitoring, considering the lack of faithfulness of CoT (Turpin et al., 2023; Chen et al., 2025; Korbak et al., 2025).

ACKNOWLEDGMENT

VS and BV were suppored by NSF CAREER Award CCF-2239265, a Google Research Scholar Award and an Okawa Foundation Research Grant. The authors acknowledge the use of USC CARC's Discovery cluster.

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

Appendix

## A  Experimental Details and Additional Experiments for Section 2

### A.1  Details on experimental setup and techniques

**Causal mediation analysis**. We primarily rely on causal mediation analysis (CMA), a.k.a. activation patching in the mechanistic interpretability literature, to obtain causal evidence for our claims in the LLM studies.

At a high level, CMA is about the study of indirect effects (IE) and direct effects (DE) in a system with causal relations (Pearl, 2022). Consider the following classical diagram of CMA, in Figure 11.

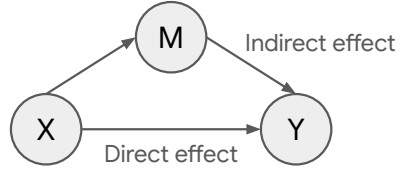

Figure 11: Basic illustration of CMA. $X$ = input (exposure), $M$ = mediator, $Y$ = output (outcome).

Suppose we wish to understand whether a certain mediator $M$ plays an important role in the causal path from the input $X$ to the outcome $Y$. We decompose the "total effect" of $X$ on $Y$ into the sum of *direct* and *indirect* effects (DEs and IEs), as shown in the figure. The indirect effect measures how important a role the mediator $M$ plays in the causal path $X \to Y$. To measure it, we compute $Y$ given $X$, except that we artificially hold $M$'s output to its "corrupted" version, which is obtained by computing $M$ on a counterfactual ("corrupted") version of the input. A significant change in $Y$ indicates a strong IE, which implies that $M$ is important in the causal path. On the other hand, a weak IE implies a strong DE, meaning that the mediator does not play a strong causal role in the system (for the distribution of inputs of interest).

There are two common classes of interventions in mechanistic interpretability for localizing model components with strong IE in the causal graph. The first class is simple ablation, such as mean ablation (replace activation of the mediator by its average output on a distribution of interest) (Wang et al., 2023) or "noising" (Meng et al., 2022). While this type of intervention is easy to perform, it typically leads to poor localization, surfacing low-level processing components irrelevant to the study (Zhang & Nanda, 2024).

The other class, which we employ, is "interchange" intervention: it requires construction of alterative prompts which differ from the normal prompt in subtle ways, requiring careful consideration of

the problem's nature, but allows "causal surgery" which surfaces model components with specific functional roles. Technically speaking, we are measuring the *natural indirect effects* of the mediator. In particular, it works as follows. We first run the system (the LLM) on both normal and alternative (or sometimes called counterfactual) inputs, and cache the output of the mediator $M$. We then hold $M$'s output to its alternative version, as we run the full system (the LLM) on the normal prompt. Everything downstream in the causal graph from $M$ are also influenced, up to the output $Y$. This helps us measure how the mediator $M$ causally implicate the answer. Or more intuitively, it measures how "flipping" the output of $M$ causally influences the LLM's "belief" in the alternative answer over the normal answer.

What makes our intervention experiments somewhat novel lies in exactly how we measure the IE. In particular, as we briefly discussed Section 2.1 and 2.2, we do *not* directly use the alternative prompt's ground truth answer to measure how well we are "bending" the model's "belief" through intervention. We discuss our method in greater detail here.

First, to understand whether certain attention heads have functional roles in processing the query source entity $S_n$ which transcend source-target types of the two-hop problems (i.e. it effectively serves as the hidden concept map $C$ as introduced in §1.1), we work with normal-alternative prompts with distinct source and target types, such as sampling an *alternative* prompt "EPFL, 41. ... University of Tokyo, " ([University, Code] problem), and a *normal* prompt "Okinawa, Tokyo. ... Chicago, " ([City, Capital] problem). We hypothesize that there are certain model components which output the bridge concept, which is then composed with the target/output concept of the problem. For the normal example, this means the *hidden* step "Chicago"→"USA" is resolved first (i.e. $C(\text{Chicago}) = \text{USA}$ here), then the model executes $\text{Capital(USA)} = \text{Washington D.C.}$ as the output (i.e. $R(\text{USA}; \text{Capital}) = \text{Washington D.C.}$). This means that, patching a model component's activation from the alternative prompt onto its activation on a normal prompt, would cause the model to favor the answer of the alternative prompt, but with the same target semantic type as the normal prompt (i.e. the map $R(\cdot; \text{Output Type})$ remains identical in its output type across the prompt pairs, but the output of $C(\cdot)$ changes). In our running example, this would be "Tokyo", the capital of "Japan", the country (bridge) of the university "University of Tokyo".

It follows that, to evaluate the "causal effects" of such a bridge-resolving component, we should set $\hat{T}_n^{(\text{alt})} = \text{Type}_{\text{norm}}(T_n^{(\text{alt})})$. We then measure the (expected) intervened logit difference

$$\Delta_{\text{alt}\to\text{norm}} = \mathbb{E}\left[\text{logit}^{\text{alt}\to\text{norm}}(\boldsymbol{p}_{\text{norm}})[T_n^{(\text{norm})}] - \text{logit}^{\text{alt}\to\text{norm}}(\boldsymbol{p}_{\text{norm}})[\hat{T}_n^{(\text{alt})}]\right], \tag{1}$$

where $\boldsymbol{p}_{\text{norm}} = [S_1^{(\text{norm})}, T_1^{(\text{norm})} ... S_n^{(\text{norm})},]$ is the normal prompt, $\text{logit}^{\text{alt}\to\text{norm}}(\boldsymbol{p}_{\text{norm}})$ indicates the logits of the model obtained after intervention while running the model on the normal prompt, and $\text{logit}(\boldsymbol{p}_{\text{norm}})$ indicates the logits of the model running naturally (un-intervened) on the normal prompt. Moreover, when we measure the rank of the model's answer when intervened, we also use $\hat{T}_n^{(\text{alt})}$ as the target.

*Remark.* To normalize our logit-difference variations, we compute

$$\bar{\Delta} = \frac{\Delta_{\text{norm}} - \Delta_{\text{alt}\to\text{norm}}}{\Delta_{\text{norm}}}, \tag{2}$$

where

$$\Delta_{\text{norm}} = \mathbb{E}\left[\text{logit}(\boldsymbol{p}_{\text{norm}})[T_n^{(\text{norm})}] - \text{logit}(\boldsymbol{p}_{\text{norm}})[\hat{T}_n^{(\text{alt})}]\right]. \tag{3}$$

**Problem settings and overall observations**. We primarily work with the Geography and Company 2-hop ICL problems to, and with the Gemma-2 LLM family. The former problem setting was described in the main text (with further elaboration in the Appendix later), while the latter will be introduced later, to add further generality to our study.

At a high level, in both the Geography and Company 2-hop ICL problems, we obtained causal and correlational evidence that highly localized groups of attention heads output the representation of the "bridge" concept based on the query, and the model later utilizes such representation to produce the output. Furthermore, an interesting technical observation is that multiplying the patched representation at these attention heads with a constant slightly above 1 (e.g. 1.5 to 4.0) tend to improve the intervention results. Finally, we make the observation that a smaller LLM, namely

Gemma-2-2B, also possesses attention heads which have a nontrivial causal role in inferring the bridge representations. However, the representations are poorly disentangled, potentially leading to the model's low accuracy on the ICL problems.

**Compute**. Our LLM experiments are conducted on 2 H200 GPUs, totaling around 200 hours of compute.

**Licenses.** We use the pretrained Gemma 2 models for our LLM experiments. They have the following license information.

- Models: Gemma-2-27B and Gemma-2-2B (google/gemma-2-27b and google/gemma-2-2B on Huggingface)
- License: Gemma Terms of Use (Google, 21 Feb 2024)
- Link: https://ai.google.dev/gemma/terms
- Notes: Commercial use permitted subject to Gemma Prohibited-Use Policy.

### A.2 Additional details on experiments with the Geography 2-hop ICL puzzles

**Dataset construction**. There are two stages to our data sampling process:

**Step 1: JSON dictionary sampling.** First, we ask ChatGPT o3 to create a JSON dictionary, mapping each country to its cities, capital, and calling code, and repeating this for universities, and famous landmarks. We then manually correct for and refine the entries in the JSON dictionaries to reduce leakage of source types. For instance, ChatGPT sometimes append city or state/province/region to a landmark, which we remove to ensure that the Landmark source type remains sufficiently distinct from the City source type. University names sometimes cannot avoid such overlap, e.g. University of California, Berkeley indeed has city name in it.

The 2-hop Company dataset is constructed in almost exactly the same fashion, mapping the companies to their products, founders and headquarter cities.

**Step 2: Prompt generation.** This is discussed in Section 2. Taking the geography puzzles as the running example, for every City→Capital prompt $X = [\text{City}_1, \text{Capital}_1 \ldots \text{City}_n,]$, we sample tuples $(\text{City}_i, \text{Country}_i, \text{Capital}_i), i = 1, \ldots, n$, where each "bridge" $B_i = \text{Country}_i$ is a different country and the $\text{City}_i$ and $\text{Capital}_i$ belong to that country, all randomly sampled from the JSON dictionary from before. The same holds for other source-target types, and on the Company dataset.

**Causal evidence**. Recall that in the main text, to provide causal evidence for the bridge-resolving mechanism, we primarily presented causal intervention experiments where we treated [City, Capital] as the problem type we intervene on, using cross-type prompts [University, Calling Code], [Landmark, Calling Code] to show causal evidence for the bridge-resolving heads. Here, we add further evidence by having other source-target types. The results are presented in Figures 12 to 21, indexed as follows:

1. Experiment [City, Capital]→[Landmark, Calling Code]: Figure 12
2. Experiment [University, Capital]→[Landmark, Calling Code]: Figure 13
3. Experiment [City, Capital]→[University, Calling Code]: Figure 14
4. Experiment [Landmark, Capital]→[University, Calling Code]: Figure 15
5. Experiment [University, Capital]→[City, Calling Code]: Figure 16
6. Experiment [Landmark, Capital]→[City, Calling Code]: Figure 17
7. Experiment [City, Calling Code]→[University, Capital]: Figure 18
8. Experiment [Landmark, Calling Code]→[University, Capital]: Figure 19
9. Experiment [City, Calling Code]→[Landmark, Capital]: Figure 20
10. Experiment [University, Calling Code]→[Landmark, Capital]: Figure 21

Every patching experiment is performed on at least 100 prompts. As we can see, the general trend is that there is strong transferability of the bridge representation across the problem types, including

when the source and target types have no overlap, giving us causal evidence that head groups (24,30;31), (35,22;23) are "resolving the bridge".

*Scaling constant for bridge intervention.* We found that for some of the transfer experiments, multiplying the patched representation for the heads (24,30;31), (35,22;23) improves the result, i.e. there is a greater percentage of samples where the alternative answer is boosted into the top-10 (or even top-1) answers of the model after intervention. Therefore, we also report those results. An intriguing property of this scaling constant is that it typically works best around 2.0. At 4.0, we often observe *saturation* or even *decline* in the intervened alternative answer's rank, such as in the [University, Capital]→[City, Calling Code] experiment shown in Figure 16.

*The output-concept heads.* While the main interest of this work lies in the bridge-resolving mechanism enabled by the sparse set of attention heads discussed above, we also present more analysis of the output-concept heads, whose embedding tends to cluster with respect to the output concept (Capital versus Calling Code). To localize these heads, we generate normal-alternative prompt pairs where we only change the target/output type of the normal prompt to generate the alternative prompt, but keep the $S_i$'s to be identical across the prompt pairs for all $i \leq n$. This helps us surface components which are independent of the query and bridge value, and sensitive to the output/target type for the ICL problem. The results are shown in Figure 22 and 23, where we run the intervention experiment [Landmark, Calling Code]→[Landmark, Capital] (due to limitations in time and computing resources, we could not sweep all the source-target combinations as of this version of the paper). As we can see, these head groups with the strongest causal scores indeed tend to exhibit sensitivity to output type, and insensitivity to source/input type and query and bridge value. Moreover, they are more concentrated in the deeper layers of the model.

**Statistics of alternative-type answers**. A natural question to challenge the bridge-resolving mechanism is as follows. Say we are performing intervention by sampling alternative prompts from the problem type [University, Calling Code], and normal prompts from the type [City, Capital]: even though the target types have little overlap in text, perhaps the model still assigns nontrivial confidence to the "Capital" version of the alternative answer (which has target type "Calling Code")? If that is so, then it challenges our hypothesis about the role of the "bridge" representations, since we might just be directly injecting the right version of the alternative answer into the model.

We show evidence to refute this. In particular, in Figure 24, we show that the model places trivial confidence on the altered-type answer, even if they share the same bridge value. Therefore, we add further evidence to the bridge-resolving mechanism.

**The role of MLPs**. We performed similar intervention experiments on the MLPs at the last token position just like with the attention heads. They are observed to primarily process the output concept type, and do not participate heavily in outputting the bridge concept representation. This is revealed in Figure 25.

**Smaller LLM exhibits weaker disentanglement**. To contrast against our results for the 27B model, we study a small model in the same family of Gemma 2 models, *Gemma-2-2B*. This smaller model has significantly lower accuracies on the problems, measured at 20 shots. [City, Capital]: 71.67%, [University, Capital]: 25.83%, [Landmark, Capital]: 66.67%, [City, Calling Code]: 47.5%, [University, Calling Code]: 57.5%, [Landmark, Calling Code]: 23.33%.

We perform an intervention experiment [University, Code]→[City, Capital] on *Gemma-2-2B*, similar to the bridge-resolving head localization experiments we did on the 27B model. Intriguingly, we were also able to surface a highly sparse set of attention heads which have nontrivial causal scores (but much lower than that achieved by the 27B model). We find that these heads exhibit noticeable, but *noisy* disentanglement with respect to the bridge representation. We show these results in Figure 26. The weaker causal score of the bridge-resolving heads and their noisier concept disentanglement in the 2B model suggests the conjecture that, the larger the model, the more specialized its concept-processing components are — assuming that the model is well-trained. Such specialization likely benefits the model's generalization accuracy.

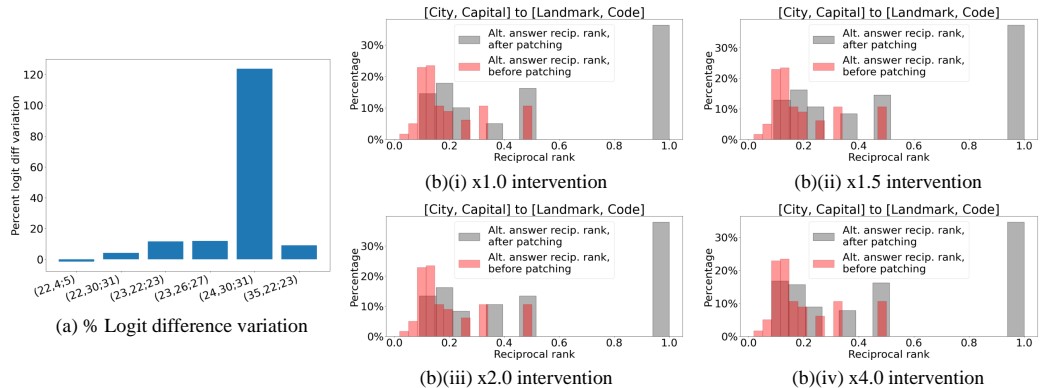

Figure 12: Gemma-2-27B [City, Capital]→[Landmark, Code] transfer experiments, intervening head groups $(24, 30; 31), (35, 22; 23)$. We show the percentage logit-difference variation in (a), and reciprocal rank of the answer answer before and after intervention in the (b) series of figures, with different scaling constants in $\{1.0, 1.5, 2.0, 4.0\}$. We observe strong causal effects of the two attention head groups. Here, the scaling constant does not significantly affect the intervention performance.

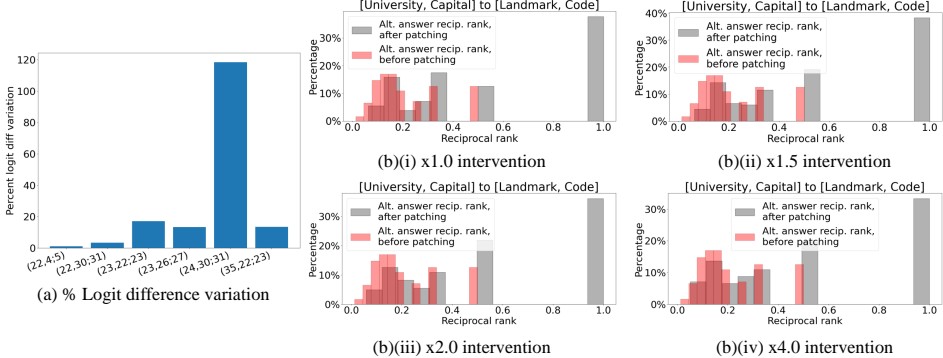

Figure 13: Gemma-2-27B [University, Capital]→[Landmark, Code] transfer experiments, intervening head groups $(24, 30; 31), (35, 22; 23)$. We show the percentage logit-difference variation in (a), and reciprocal rank of the answer answer before and after intervention in the (b) series of figures, with different scaling constants in $\{1.0, 1.5, 2.0, 4.0\}$. We observe strong causal effects of the two head groups. Interesting, past a scaling constant of 1.5, we observe decline in intervention performance.

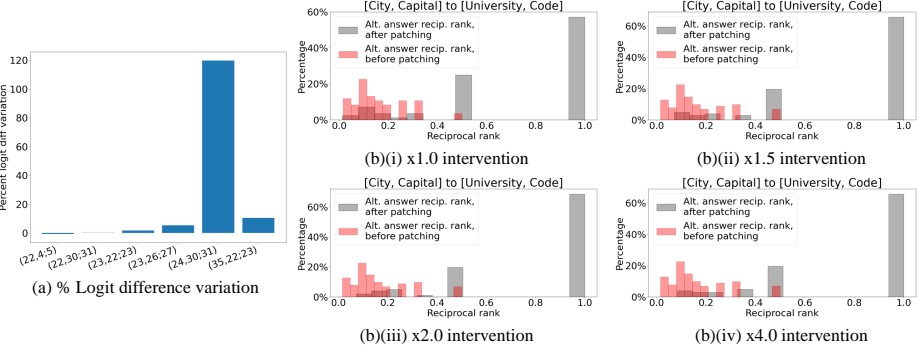

Figure 14: Gemma-2-27B [City, Capital]→[University, Code] transfer experiments, intervening head groups $(24, 30; 31), (35, 22; 23)$. We show the percentage logit-difference variation in (a), and reciprocal rank of the answer answer before and after intervention in the (b) series of figures, with different scaling constants in $\{1.0, 1.5, 2.0, 4.0\}$. We observe strong causal effects of the two attention head groups, boosting the alternative answer into top 1 around 60% of the time! Additionally, the scaling constant does not significantly affect the intervention performance in this experiment.

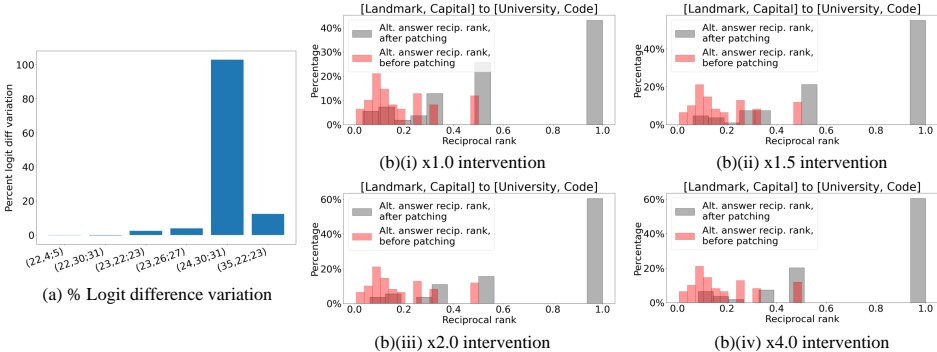

Figure 15: Gemma-2-27B [Landmark, Capital]→[University, Code] transfer experiments, intervening head groups $(24, 30; 31), (35, 22; 23)$. We show the percentage logit-difference variation in (a), and reciprocal rank of the answer answer before and after intervention in the (b) series of figures, with different scaling constants in $\{1.0, 1.5, 2.0, 4.0\}$. We observe strong causal effects of the two attention head groups. Here, the positive effects of the scaling constant saturates around 2.0.

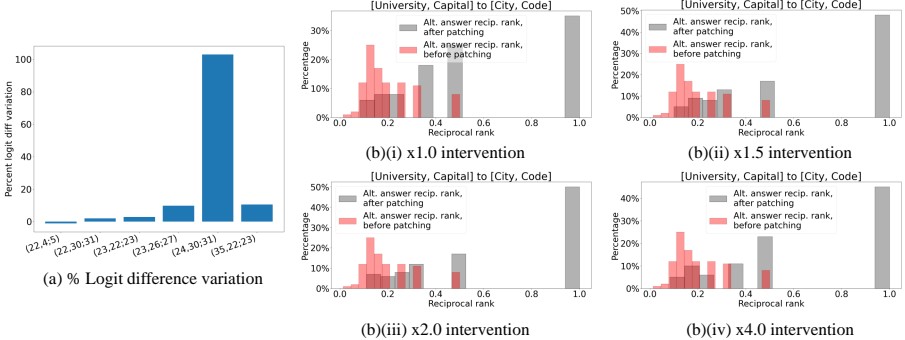

Figure 16: Gemma-2-27B [University, Capital]→[City, Code] transfer experiments, intervening head groups $(24, 30; 31), (35, 22; 23)$. We show the percentage logit-difference variation in (a), and reciprocal rank of the answer answer before and after intervention in the (b) series of figures, with different scaling constants in $\{1.0, 1.5, 2.0, 4.0\}$. Interestingly, we observe decline in the intervention's accuracy as we push the scaling constant from 2.0 to 4.0 (top-1 accuracy decreases from around 50% to slightly above 40%), indicating a subtle regime in which the scaling constant boosts intervention performance.

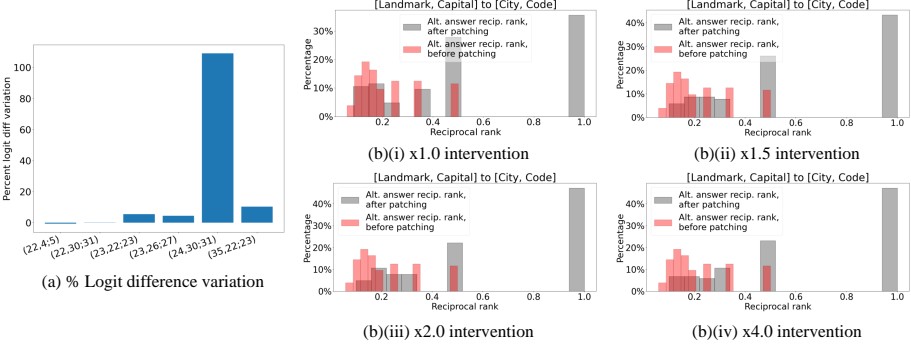

Figure 17: Gemma-2-27B [Landmark, Capital]→[City, Code] transfer experiments, intervening head groups $(24, 30; 31), (35, 22; 23)$. We show the percentage logit-difference variation in (a), and reciprocal rank of the answer answer before and after intervention in the (b) series of figures, with different scaling constants in $\{1.0, 1.5, 2.0, 4.0\}$. We observe strong causal effects of the two attention head groups.

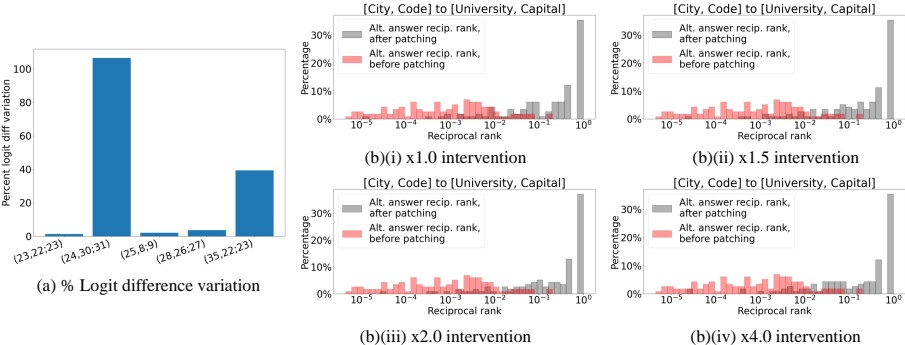

Figure 18: Gemma-2-27B [City, Calling Code]→[University, Capital] transfer experiments, intervening head groups $(24, 30; 31), (35, 22; 23)$. At scaling constant 1.0 (i.e. natural intervention, no additional scaling), the reciprocal rank of 0.1 for the alternative answer after intervention is at the $36^{th}$ percentile, while before patching, as we can see, the reciprocal rank of the alternative answer is mainly in the range of $10^{-2}$ to $10^{-5}$.

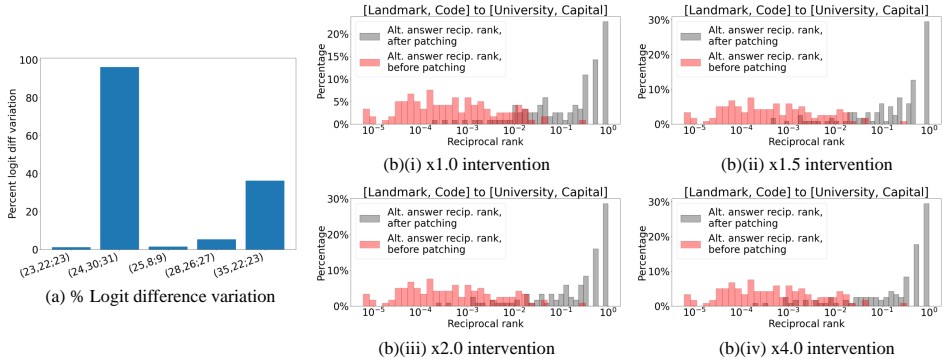

Figure 19: Gemma-2-27B [Landmark, Calling Code]→[University, Capital] transfer experiments, intervening head groups $(24, 30; 31), (35, 22; 23)$. At scaling constant 1.0 (i.e. natural intervention, no additional scaling), the reciprocal rank of 0.1 for the alternative answer after intervention is at the $41^{th}$ percentile.

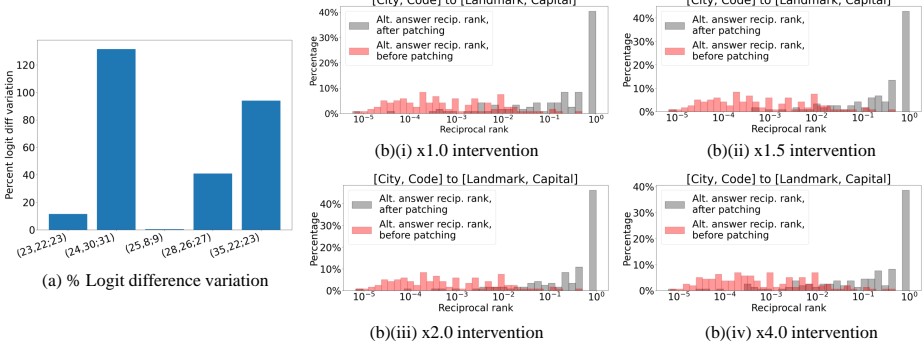

Figure 20: Gemma-2-27B [City, Calling Code]→[Landmark, Capital] transfer experiments, intervening the single head group $(24, 30; 31)$. At scaling constant 1.0 (i.e. natural intervention, no additional scaling), the reciprocal rank of 0.1 for the alternative answer after intervention is at the $34^{th}$ percentile.

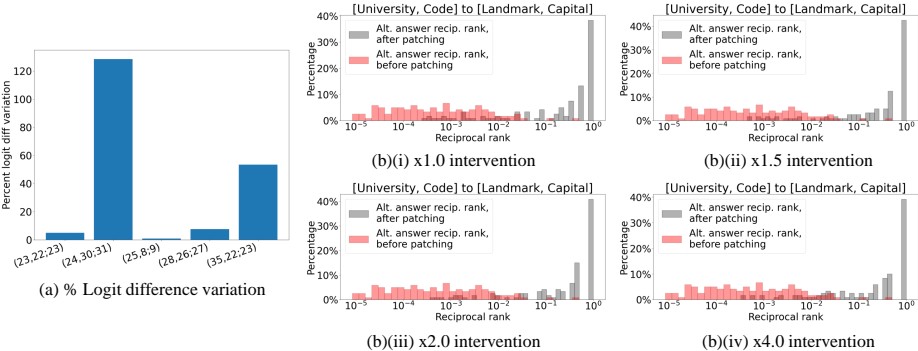

Figure 21: Gemma-2-27B [University, Calling Code]→[Landmark, Capital] transfer experiments, intervening the single head group $(24, 30; 31)$. At scaling constant $1.0$ (i.e. natural intervention, no additional scaling), the reciprocal rank of $0.1$ for the alternative answer after intervention is at the $31^{st}$ percentile.

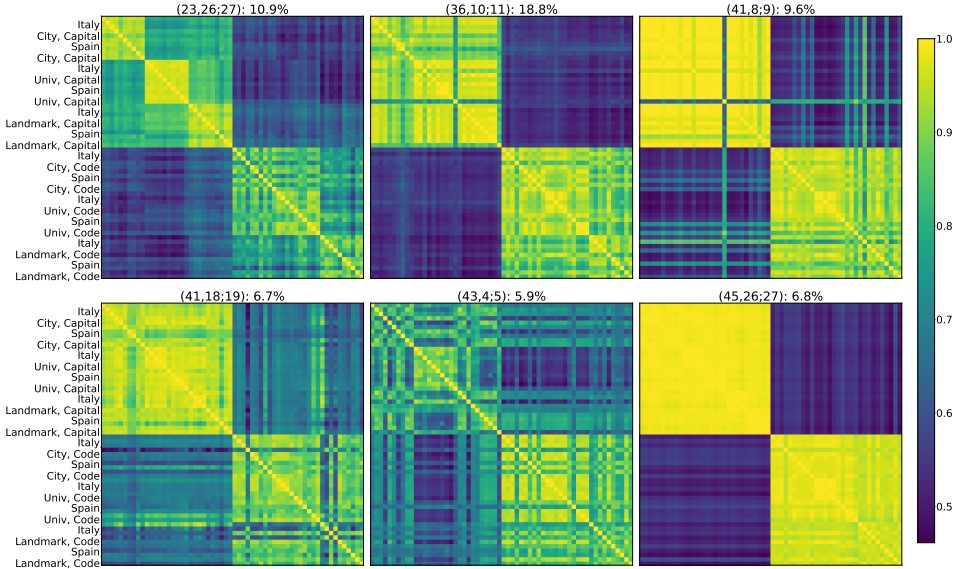

Figure 22: (Best viewed zoomed in) Cosine similarity map of the output-concept head groups (with top causal scores) identified in Gemma-2-27B, along with the percent logit-difference variation of the head groups, serving as the metric for the head groups' causal effects. Observe that they are mostly insensitive to the source type, query value, and bridge value, and primarily sensitive to the output/target type. Note: in this set of visualizations, we are using "Italy" and "Spain" as the bridge values.

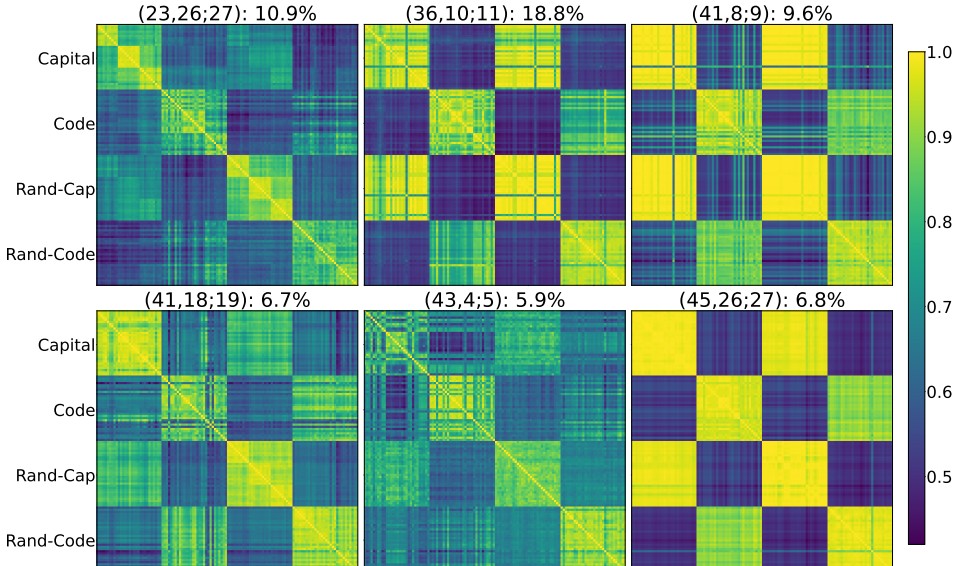

Figure 23: Cosine similarity map of the output-concept head groups identified in Gemma-2-27B. Here, we construct four groups of prompts. The first two groups consist of multi-hop ICL problems with Capital or Calling Code as the target type. The remaining two are created by randomly shuffling the output of the normal multi-hop ICL samples, causing the problem to essentially demand randomly outputting a Capital or a Calling Code; these are the "negative controls" we discussed in the main text. We find the output-concept heads' embeddings on the multi-hop prompts to align strongly with those on the output-concept-only prompts, further confirming their role in the circuit.

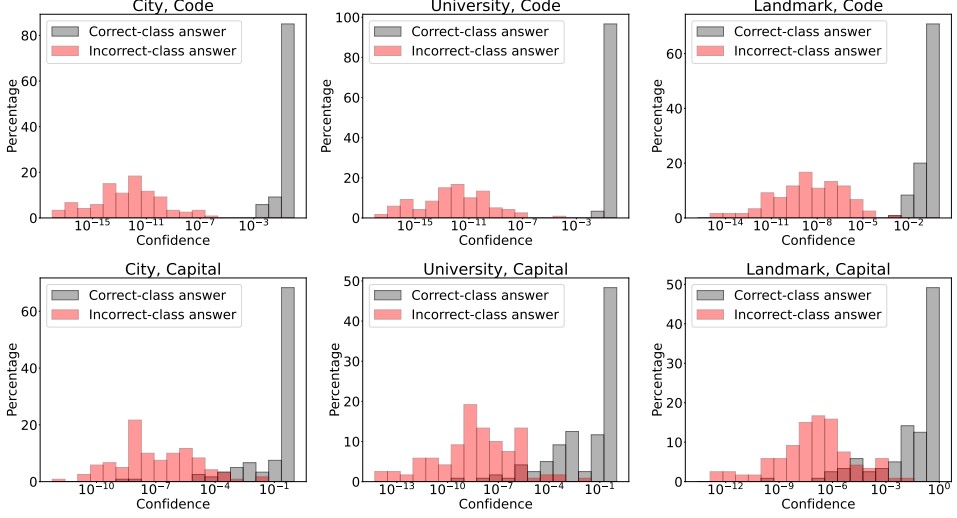

Figure 24: Distribution of Gemma-2-27B's confidence on the correct- and incorrect-type answer on the different problems. When we say "correct-class" answer, we simply mean that the answer's semantic type aligns with that of the problem's target, e.g. the correct-type answer for a prompt "Okinawa, Tokyo. Nantes, Paris. ... Shanghai, " would be "Beijing", while the incorrect-type answer would be "86" (the calling code of China, which the city Shanghai belongs to). We observe a clear separation in the LLM's confidence between the two types of answers.

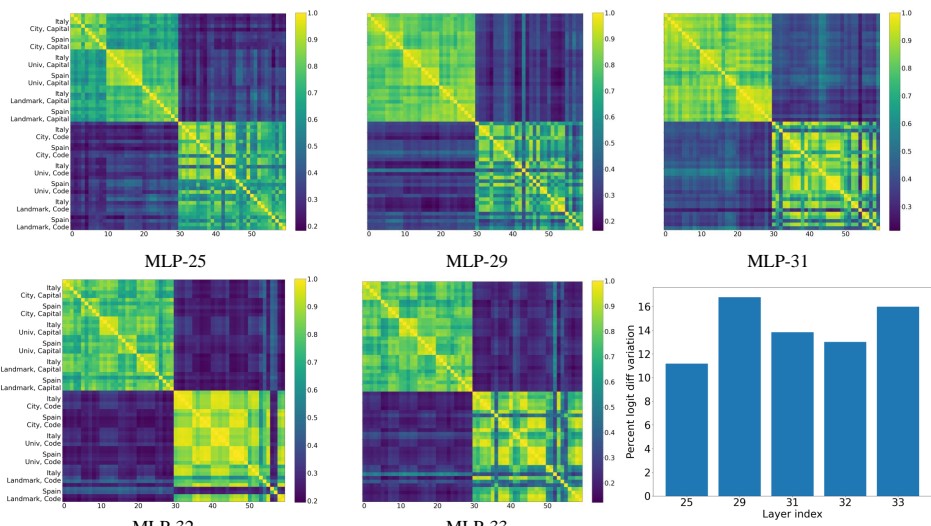

Figure 25: Functional roles of the MLPs in Gemma-2-27B. We perform patching experiments on [University, Code]→[City, Capital] at the last token position, similar to how we localize the bridge-resolving attention heads. We report the percentage logit-difference variation of the top-scoring MLPs, along with their cosine similarity maps computed on prompts sampled with different combinations of bridge values ("Italy" and "Spain") and diverse set of source-target types. Perhaps unsurprisingly, the MLPs at the last token position play a less interesting role: as seen in the cosine similarity maps for the MLPs with the highest causal scores (fairly low compared to the attention heads), they primarily discriminate against the output type. They do not appear to participate much in resolving the bridge concept, as indicated by their lower causal scores, and the lack of sensitivity to the bridge entities in the cosine-similarity plots.

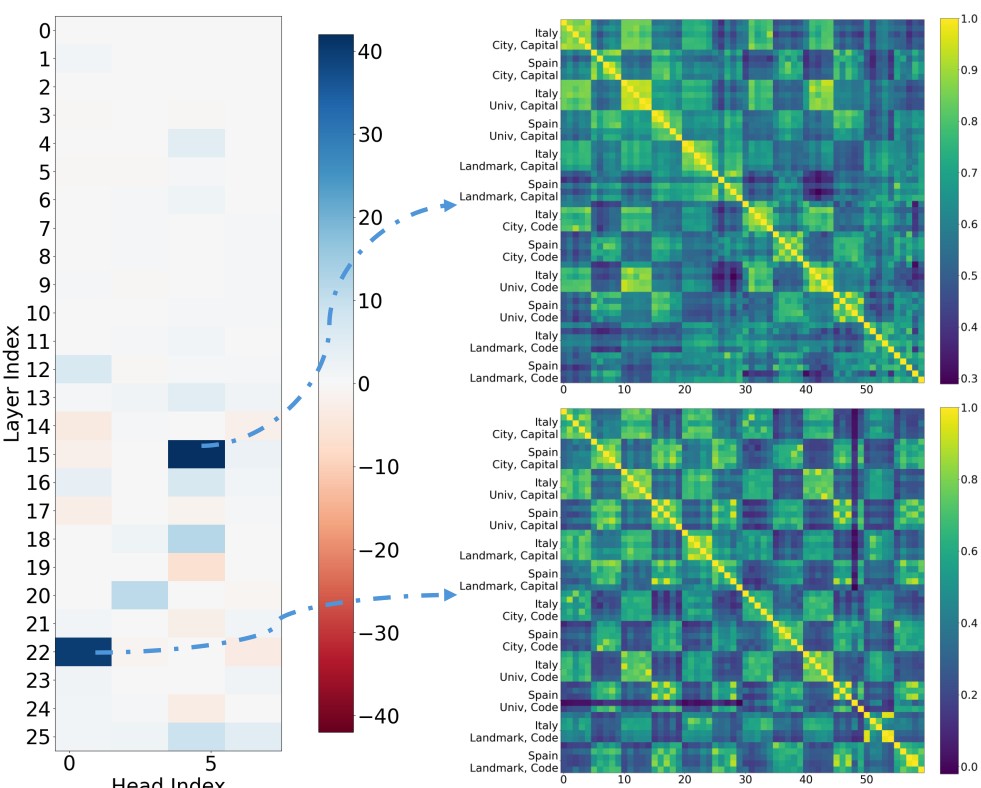

Figure 26: Results of the intervention experiment [University, Code]→[City, Capital], conducted on Gemma-2-2B, with 20-shot ICL. On the left, we show the percentage logit difference variation of the intervention experiment; on the right, we plot the cosine similarity map of the two head groups with the highest causal scores, namely $(15, 4; 5)$ and $(22, 0; 1)$. We find that while the two attention head groups exhibit nontrivial causal effects and disentanglement, they are, in comparison, much weaker than those exhibited by the 27B model. This likely explains the significantly lower accuracy of the 2B model than the 27B model. This also suggests a conjecture: perhaps the larger the model, the more specialized its concept-processing components are?

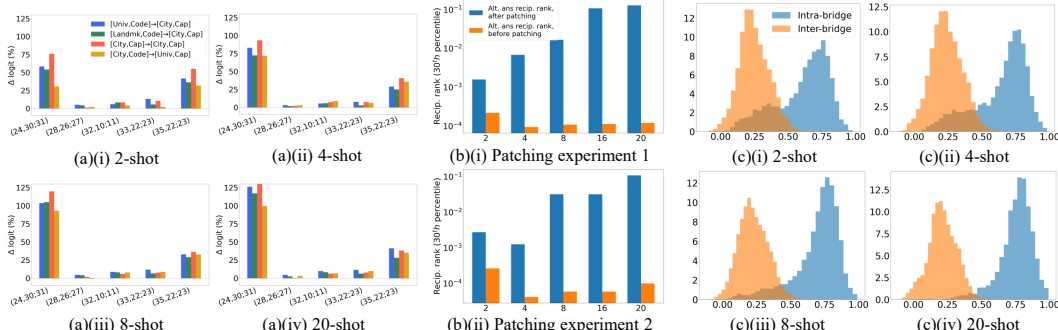

Figure 27: This figure illustrates how the transferability and disentanglement of the bridge representation increases as we increase the number of in-context examples. Figure series (a) and (b) present the transferability result, obtained by performing cross-problem-type patching, and measuring the causal influence of the patched representation. In (a)(i) to (iv), we plot the percentage logit variation of the attention heads found to output "bridge" values, measured on several intervention experiments. For (b)(i) and (ii), we zoom in on head group (24,30;31), and show its causal effects on two patching experiments, [University, Calling Code]→[City, Capital] and [Landmark, Calling Code]→[City, Capital]. The x-axis is the number of in-context examples, and the y-axis is the $30^{th}$ percentile of the reciprocal rank of the alternative prompt's answer. For (c)(i) to (iv), we plot the histogram of the disentanglement strength of the representations of head group (24,30;31), with the y-axis in percentage, and x-axis being cosine similarity.

## A.3 MULTI-HOP MECHANISM FORMATION AND THE NUMBER OF IN-CONTEXT EXAMPLES

This sub-section focuses on illustrating the relation between the number of in-context examples versus (1) how strong a role the multi-hop mechanism plays in the LLM's inference (via causal interventions), (2) disentanglement strength of key bridge-resolving attention heads. As we will show below, there is a general positive correlation between the number of shots and the two factors.

**More demonstrations $\implies$ stronger causal score**. Figure 27 visualizes the experimental results. From Figure 27(a) and (b) and sub-figures, we observe a correlation between the number of shots (ICL examples) and the bridge-resolving heads' "causal importance" in the model's inference. When the number of shots is low, we find that they tend to exhibit weak causal influence on the model's inference. For instance, as (b)(i) shows, at 2 shots, the $30^{th}$ percentile of the alternative answer's rank *after* patching at (24,30;31) is on the order of $10^3$. This is in stark contrast to how strong this head group's causal influence is at 20 shots as we saw before.

**More demonstrations $\implies$ stronger disentanglement, with a catch**. In Figure 27(c)(i) to (iv), we observe that the intra-bridge cosine similarity tends to cluster better as the number of shots increase, while the inter-bridge cosine similarities decay toward 0.2, with the two distributions overlapping less and less. Interestingly, the bridge-disentanglement strength is still non-trivial with very few shots, mirroring the causal-intervention results: regardless of how disentangled the representations are in the very-few-shot regime, the LLM does not "realize" how it should utilize the multi-hop sub-circuit.

## A.4 PRELIMINARY EVIDENCE OF CAUSAL TRANSFERABILITY OF CONCEPT EMBEDDINGS FROM ICL PUZZLES TO NATURALISTIC CONTEXTS

**This subsection was added in the Rebuttal & Discussion Phase.**

A natural question regarding the concept embeddings found in the synthetic ICL-puzzle setting stands: Are these "concept embeddings" specific to those strictly formatted puzzle-like tasks, or are they reflect mechanisms that transfer to more naturalistic settings? Are these embeddings merely artifacts of that artificial setup?

While fully resolving this question is beyond the scope of this work, we conduct a small-scale study that provide preliminary but positive evidence that, the concept embeddings found in the ICL-puzzle setting are not mere artifacts of the artificial setup: they exhibit causal influence on generation in more naturalistic prompts.

**Experimental setup.**

*Naturalistic context prompts.* We continue to work with Gemma-2-27B. We extract "Country" concept embeddings solely from our synthetic ICL puzzles (as described in Section 2), using only the City→Capital source-target type. To test their causal transferability to naturalistic settings, we inject these vectors at attention layer 24 (as localized in the ICL setting) when the model is given open-ended, natural-sentence prompts.

The natural prompts are still controlled, and constructed as follows:

- A short *"hint" sentence* that implicitly describes a country, e.g., "From cherry blossoms to temples, everything here is unique." (Hidden concept = Japan)

- A *"Type" sentence*, with Type $\in$ {Universities, Celebrities, Tourist Spots}, e.g., "I also know of some celebrities there, such as" (Type = Celebrities)

An example prompt is:

```
From cherry blossoms to temples, everything here is unique.
I also visited some great tourist spots there, such as
```

We focus on a small set of target countries (corresponding to the concept embeddings which we inject into the model) with a strong presence on the internet (US, UK, China, South Korea, India) for the sake of tractability.

*Causal transfer experiment.* We perform the following causal transfer experiment. First, we extract the Country concept embeddings from the ICL City→Capital setting, by *averaging* Attention Layer 24's activation[6] at the last token position (with the query belonging to a chosen country) over 50 prompts, with 16 in-context examples for each prompt. We denote a Country concept embedding as $c_{\text{country}}$. Then, as we run Gemma-2-27B on the naturalistic context prompt, at *every* token position (including the *already generated* token positions), for Attention Layer 24 activation, we perform the following intervention

$$a_{\ell=24} \leftarrow 0.5\, a_{\ell=24} + \gamma\, c_{\text{country}} \tag{4}$$

where $\gamma$ denotes the scaling constant, which is set to $3.0$ in our experiments – this scaling strength was observed to work well in the majority of our ICL-puzzle experiments, so we did not perform ablations on it, considering resource limitations. As we rely on the `transformers` library (Wolf et al., 2020) for our experiments here, for text generation, we set the temperature to $0.7$, and `top_p` to $0.9$ (we only sample from the smallest set of tokens whose cumulative probability exceeds $0.9$), which are fairly standard hyperparameters to use.

We then use an LLM judge (gpt-4o-mini) to evaluate whether the intervention causes the continuation to (i) switch from the source country implied by the context to the target country encoded by the concept vector, and (ii) remain specific to the chosen Type. The judge returns a relevance score from 1 (irrelevant to the target country/Type) to 5 (highly relevant). We then manually correct these scores

---

[6]We work with the post-RMSNorm attention layer activation in the Gemma 2 model, namely `post_attention_layernorm` in the `transfomers` API.

(we work with a total of around 100 prompts, ensuring tractability of manual inspection). We will elaborate further on our use of gpt-4o-mini later in this subsection.

**Results: causal influence in naturalistic prompts**.

We obtain the following quantitative results.

- *Pre-intervention*: Continuations naturally stay in the source country, with average target-country relevance approximately 1.0;

- *Post-intervention*: For a substantial fraction of cases (roughly 70% for Celebrities Type, and around or above 80% for the Tourist Spots and Universities Types), the 27B model's continuations are judged as describing the target country while remaining on the specified Type.

Please refer to Figure 28 for the visualized results.

Importantly, the generations remain coherent; the interventions do not simply inject country tokens, but produce fluent text that *integrates the target concept within the given Type*. We present a list of examples in Table 1.

These results suggest that the *concept embeddings discovered in the ICL puzzle setting* behave as meaningful control directions for country-level content in more naturalistic text rather than being tied to our synthetic format, and are, at least in our settings, *disentangled from superficial linguistic patterns*.

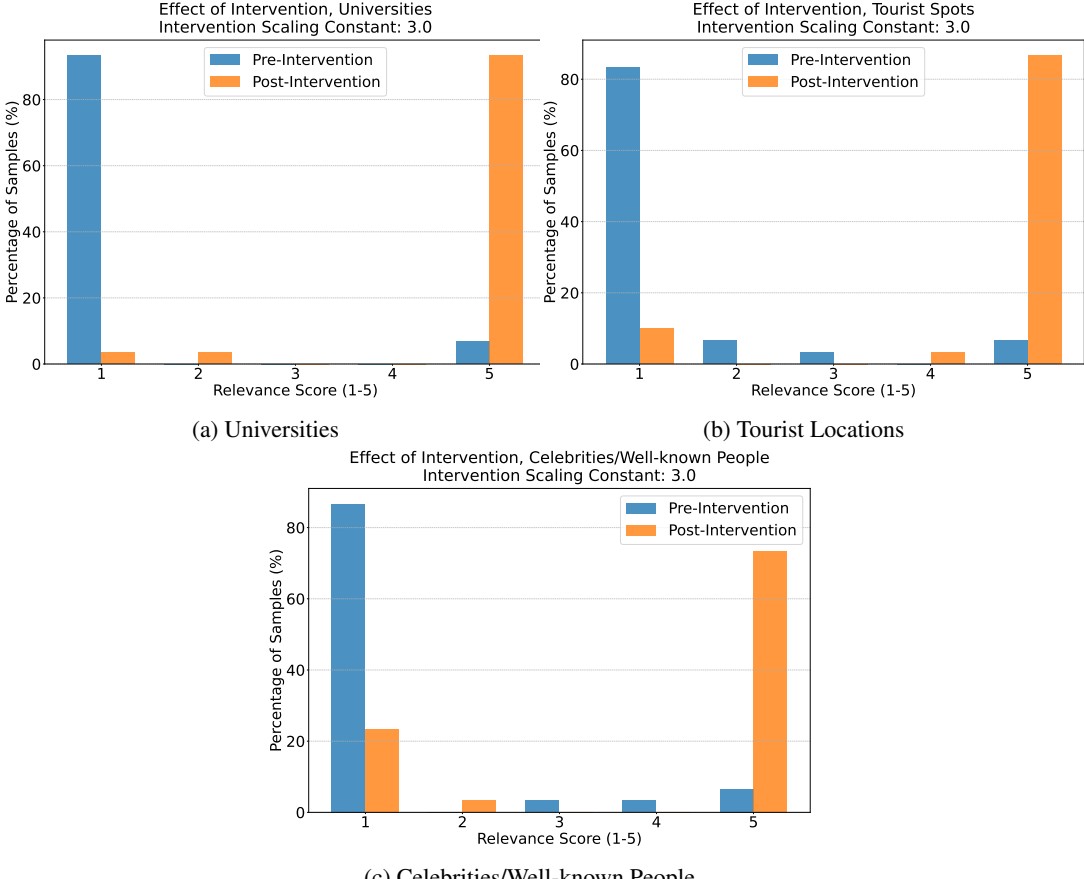

(a) Universities

(b) Tourist Locations

(c) Celebrities/Well-known People

Figure 28: LLM judge scores (with manual corrections) with and without intervention at Attention Layer 24 of *Gemma-2-27B*, using the concept embeddings we extracted from the ICL puzzles limited to source-target type City→Capital. We observe consistent shift toward the injected country in Gemma-2-27B's text continuation while correctly maintaining the Type from the context prompt, for all three Type ∈ { Universities, Tourist Spots, Celebrities/Well-known People}.

Table 1: Examples of context prompts, natural and intervened continuations of Gemma-2-27B.

| Problem info; Type | Context Prompt | Natural Continuation | Intervened Continuation |
|---|---|---|---|
| Japan to South Korea; Tourist Spots | From cherry blossoms to temples, everything here is unique. I also visited some great tourist spots there, such as | the Tokyo Tower and the Tokyo Imperial Palace. I also visited the Meiji Shrine, which was beautiful and | the Gyeongbokgung Palace, the Namsan Tower, and the N Seoul Tower. |
| South Korea to US; Tourist Spots | From K-pop to technology, everything here is unique. I also visited some great tourist spots there, such as | the Namsan Tower, the Gyeongbokgung Palace and the Namsan Park. | Disney World, Universal Studios, and the Statue of Liberty. The United States of America is a country |
| China to South Korea; Tourist Spots | I loved everything about my trip - the high-speed rail, the tea culture, and especially the lanterns. I also visited some great tourist spots there, such as | the 99 Dragons Wall, the Xiamen University campus, and the Huandao Road. | the Jeju Olle Trail, Seongsan Sunrise Peak, and Udo Island. I really |
| China to UK; Celebrities | From lanterns to tea culture, everything here is unique. I also know of some celebrities there, such as | Zhang Yimou and Gong Li. I love the food there, like hot pot and dumplings. | George Harrison, who visited this place. It was the time when The Beatles were rising to stardom |
| India to China; Celebrities | The cricket was amazing, and you can't find rickshaws like this anywhere else. I also know of some celebrities there, such as | a young man named Asif. I'm happy to see him now. A very good | the great Zhang Guorong, who used to live here. The most famous restaurant in the area |
| Canada to India; Celebrities | From poutine to moose, everything here is unique. I also know of some celebrities there, such as | the famous singer, Celine Dion, who was born and raised in Montreal. If you are looking for | Amitabh Bachchan and Salman Khan. <h2>Is India a good country to visit?</h2> |
| France to US; Universities | I loved everything about my trip - the croissants, the Eiffel Tower, and especially the baguette. I also visited some universities there, such as | Sciences Po and the Paris Institute of Political Studies. I was really impressed by the quality of education | Harvard and Yale. Just kidding. I'm not going to write about France. I'm |
| India to UK; Universities | The Bollywood was amazing, and you can't find rickshaws like this anywhere else. I also visited some universities there, such as | the one where the Bollywood movie "3 Idiots" was filmed. I've heard a | Cambridge University. It was a very different experience. It's a bit like a scene from |
| US to China; Universities | I loved everything about my trip - the bald eagle, the Hollywood, and especially the fast food. I also visited some universities there, such as | UCLA and UC Berkeley. I had a great time with my friends and the people there were so friendly | Beijing University, Tsinghua University, and others. I got to see the Chinese people's |

**Scaling (model size and disentanglement)**. We also repeated the same intervention experiments on Gemma-2-2B (intervening at layer 15, which showed weaker but non-trivial causal scores in our ICL experiments before; see Figure 26). As shown in Figure 29, the fraction of post-intervention continuations receiving score 5 from the judge is roughly 1.5 to 2 times lower for the 2B model than for the 27B model. While still preliminary, this is consistent with our ICL-setting findings and supports the interpretation that concept disentanglement and transferability improve with model scale.

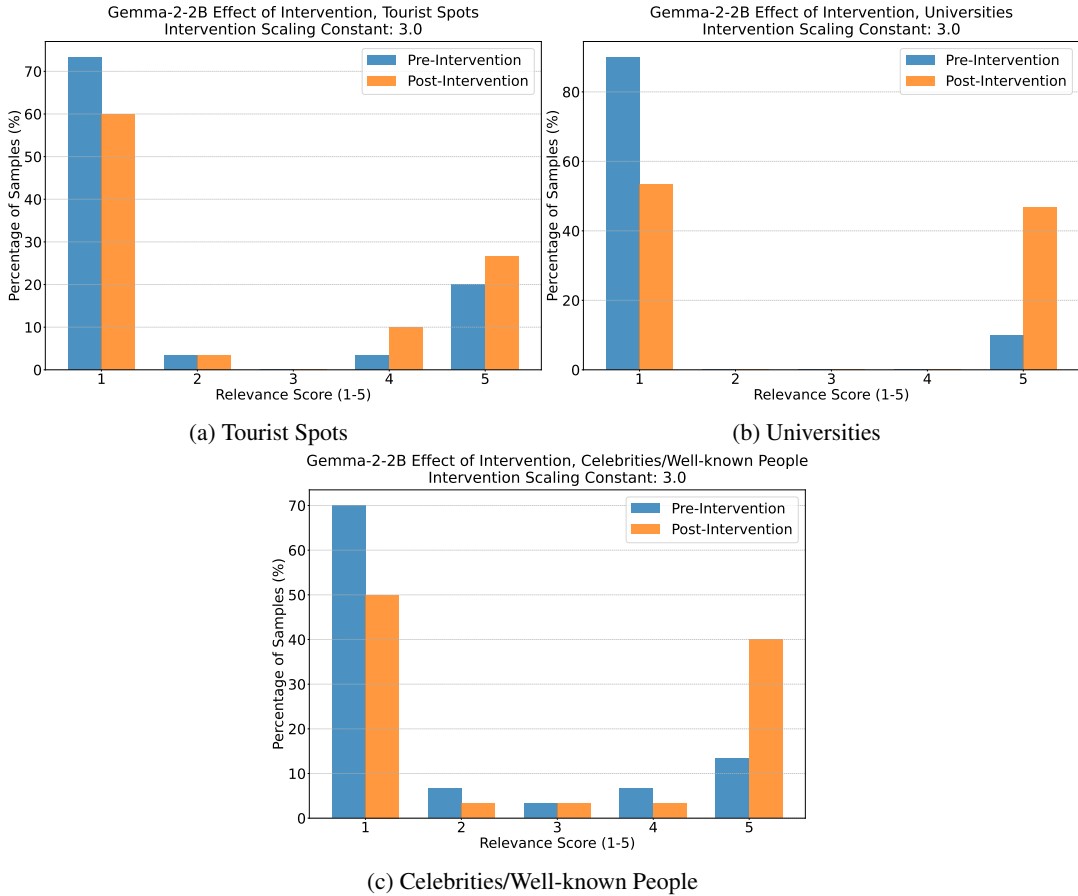

(a) Tourist Spots                     (b) Universities

(c) Celebrities/Well-known People

Figure 29: LLM judge scores (with manual corrections) with and without intervention at Attention Layer 15 of *Gemma-2-2B*, using the concept embeddings we extracted from the ICL puzzles limited to source-target type City→Capital. We observe much weaker shift toward the injected country in the text continuation of the context prompt, compared to the 27B's results.

**LLM as judge (GPT-4o-mini)**. We use the following prompt to query GPT-4o-mini (OpenAI, 2024) to score the LLM's text generation.

```
Analyze the following text and determine if it relates to the
{target_type} of {target_country}.

TEXT: "{text}"

INSTRUCTIONS:
1. Identify any specific entities (people, places, landmarks)
in the text.
2. For each geographic location, explicitly state their real-world
country of origin/location.
3. For people, explicitly state their real-world country
or countries of origin/location.
4. Compare these locations to the target country: {target_country}.
5. Based ONLY on this comparison, assign a relevance score (1-5).

SCORING CRITERIA:
1: No relation or talks about a different country.
2-3: Vague connection or generic sentences.
4-5: Clearly mentions about the {target_type} of {target_country}.

FORMAT:
Reasoning: [Your step-by-step analysis here]
Score: [Integer only]
```

where `text` refers to the model's generation, `target_type` refers to Type in {Universities, Tourist Spots, Celebrities or well-known people}, and `target country` refers to the country which the injected concept embedding corresponds to.

Additionally, we found that without allowing for some reasoning/chain-of-thought token, GPT-4o-mini tends to hallucinate in classifying the model's continuations, especially for the Celebrities type. We set 150 max tokens of chain-of-thought, which was observed to significantly reduce hallucinations.

### A.5   THE COMPANY PUZZLES, ANOTHER MULTI-HOP ICL TASK

**Problem setup**. To complement our study on the Geography puzzles and for more generality of our conclusion that sufficiently capable LLMs solve simple 2-hop ICL problems by resolving the bridge entities first, we work with another set of problems on globally well-known companies in the world. We collect data on 36 of them.

Here, our evidence is provided on a narrower range of source and target entity types, due to the nature of the problem. In particular, we allow source entities to fall in {Products, Founders}, and target entities to fall into {Founders, Headquarters City}. Furthermore, to obtain causal evidence that the model is resolving the bridge values from the query before producing the actual answer, we perform patching experiments with [Products, Founders]→[Founders, HQ City] and [Founders, Products]→[Products, HQ City]. If we again observe that there is high rank of the alternative prompt's *converted* answer after patching a sparse set of attention heads on the final token position, then we see significant causal evidence that the model resolves the bridge value during inference.

Some manual cleaning was needed in creating this dataset, on top of ChatGPT sampling: we primarily filter out products which had company names in them, and companies whose products mostly contain the company name. An example category of this would be banks and credit/debit card services, e.g. Visa, Mastercard, HSBC, etc. For companies with multiple headquarters, we choose the most well-known one out of them as the correct "Headquarter City" of the company; this sometimes counts an old headquarter of the company as the correct one. *Note that this causes us to under-estimate the LLM's accuracy with and without intervention!*

**Quantitative results**. We primarily work with the 16-shot ICL setting due to limited compute. In this setting, the [Product, Founder], [Product, HQ City] and [Founder, HQ City] have accuracies $73.5\%, 80.7\%, 88.6\%$ respectively. We wish to note that this set of puzzles does rely on significantly more obscure knowledge than the Geography puzzles, so lower accuracies and weaker multi-hop links are expected overall.[7]

As Figures 30, 31 and 32 show, we again obtain evidence for the bridge-resolving heads. In particular, we find that head groups $(22, 2; 3), (24, 14; 15), (24, 30; 31), (28, 12; 13), (28, 26; 27), (35, 22; 23)$ ($0.8\%$ of the total number of attention heads) exhibit the strongest causal effects in our experiments, and their output representations exhibit strong disentanglement with respect to the companies.

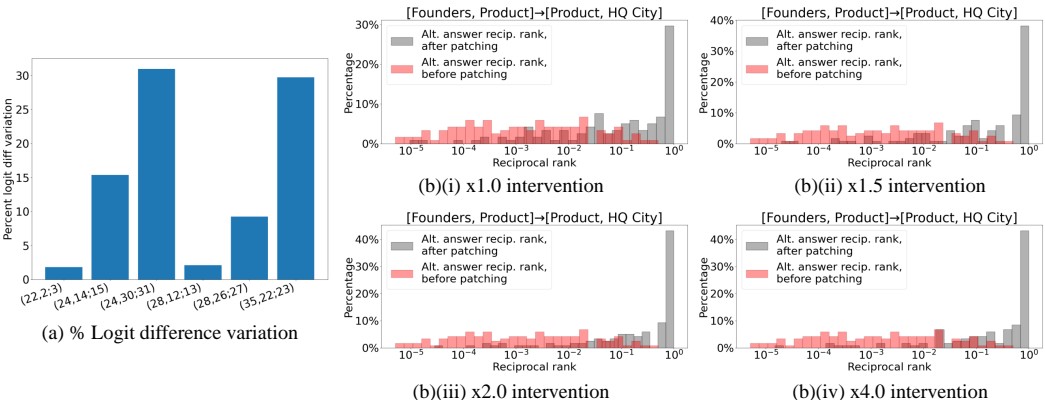

(a) % Logit difference variation

(b)(i) x1.0 intervention

(b)(ii) x1.5 intervention

(b)(iii) x2.0 intervention

(b)(iv) x4.0 intervention

Figure 30: [Founder, Product]→[Product, HQ City] transfer experiments, intervening the head groups $(22, 2; 3), (24, 14; 15), (24, 30; 31), (28, 12; 13), (28, 26; 27), (35, 22; 23)$ ($0.8\%$ of the total number of attention heads).

---

[7]We suspect that lower accuracy is not only due to the more obscure entities in this problem, but also because of several ambiguities. For example, (1) Product names are oftentimes simply not presented alongside company names, e.g. "Gatorade" is often not written alongside "PepsiCo"; (2) There can be confusion between company founders and well-known product directors, especially in certain art/entertainment companies.

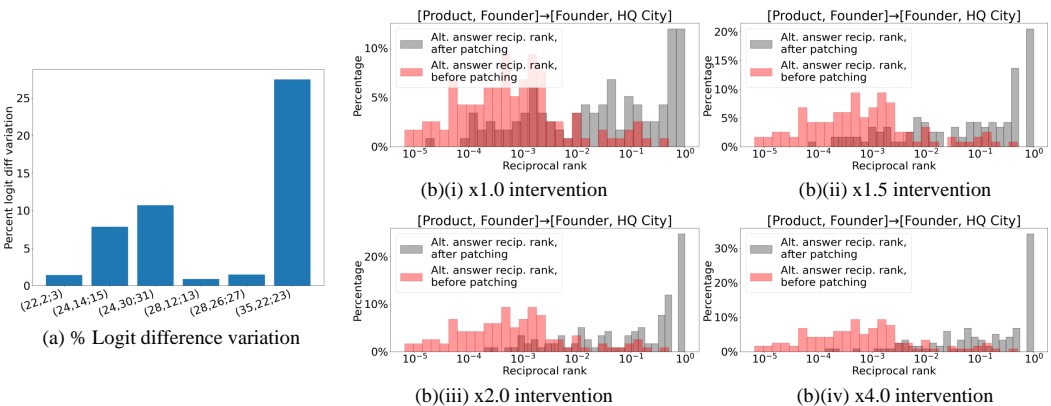

Figure 31: [Product, Founder]→[Founder, HQ City] transfer experiments, intervening the head groups $(22, 2; 3), (24, 14; 15), (24, 30; 31), (28, 12; 13), (28, 26; 27), (35, 22; 23)$ $(0.8\%$ of the total number of attention heads).

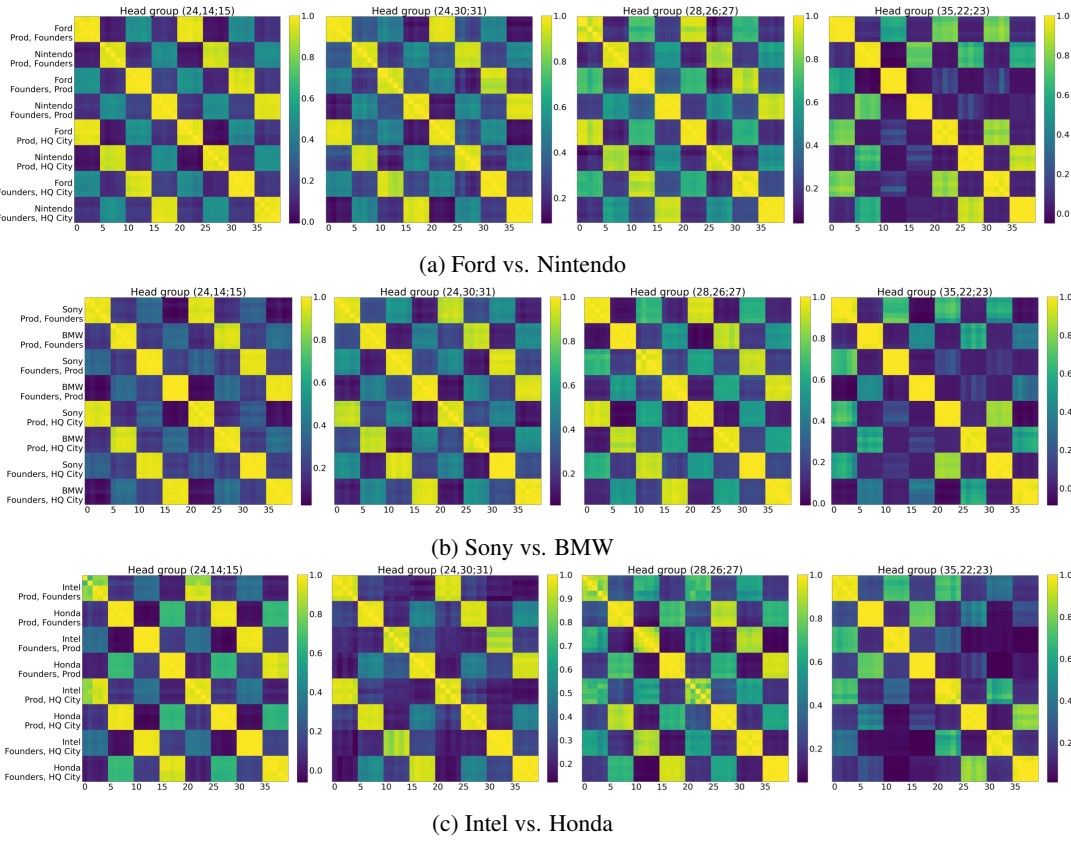

Figure 32: Cosine similarity of representations from the top-4 attention heads for resolving the bridge entity in Company ICL problems. For each combination of Bridge choice and source-target type, we sample 5 prompts to compare representation similarity on. We notice that the heads exhibit disentanglement with respect to the bridge concepts (the companies), but individually, they are not as capable in clearly resolving the bridge concepts in the Company problems as they do on the easier Geography problems (the latter problem having relevant accessible knowledge on many more websites than the former). This likely explains the model's lower accuracies and somewhat weaker intervention results on the Company problems, compared to the Geography problems.

# B    EXPERIMENTAL DETAILS AND ADDITIONAL EXPERIMENTS FOR SECTION 3

This section includes experimental details and additional experiments for problems with numerical or continuous parameterization considered in Section 3.

**Compute.**    The experiments in this section were performed on an internal cluster with a P100 and a V100 GPU. We list training details such as batch sizes and iterations used for the experiments in their respective sections.

## B.1    MORE ON THE ADD-$k$ PROBLEM

In this section, we investigate the effect of increasing the number of tasks on the results shown in Fig. 8, where we showed that the task vectors for $K = 4, 8, 16$ offsets lie on a 1D linear manifold. In these settings, the model reached $\approx 100\%$ train and test accuracy.

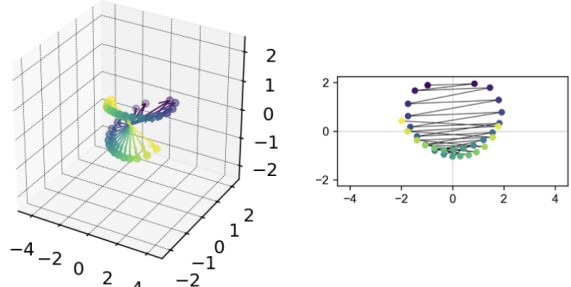

We consider a setting with 32 offsets with gap between the offsets $k_{i+1} - k_i = 1$. In this setting, the train/test accuracies reach only about $90\%$, which indicates this is a harder problem for the model to solve. Fig. 33 shows 3D and 2D PCA projections of the task vectors. These explain about $96.1\%$ and $84.48\%$ of the variance, respectively. We see that the task vectors lie on a 3D manifold that looks like a (small) helix, with alternating (odd and even) offsets represented in two separate arcs of the helix. This is significantly different from the 1D linear manifold we observed with a smaller number of offsets in Fig. 8. This type of geometry is reminiscent of the observations in Zhou et al. (2024); Kantamneni & Tegmark (2025), which study how pre-trained language models do addition and find that these models encode numbers as a helix.

Figure 33: 3D (left) and 2D (right) PCA projections of the task vectors for the *add-k* problem with 32 offsets. The task vectors lie on a 3D manifold that looks like a (small) helix. From the 2D projection, we see that alternating (odd and even) offsets are represented in two separate arcs of the helix.

**Training Details.**    The models were trained for 1000 iterations using AdamW optimizer with learning rate 0.001 and weight decay 0.01. We use a linear learning-rate schedule with a warm-up phase over the first $10\%$ iterations. The train data contains $5000K$ sequences, we use a batch size of 500 for $K = 4, 8, 16$, and 2000 for $K = 32$.

## B.2    MORE ON THE CIRCULAR-TRAJECTORY PROBLEM

In this section, we include further details about the experimental setup of the Circular-Trajectory problem, and present some additional results.

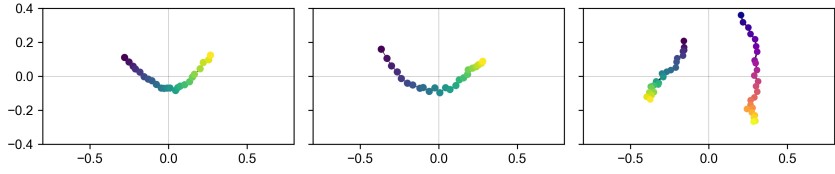

Figure 34: 2D projection of task vectors for the Circular-Trajectory problem when the model is trained on 32 radius values. The three plots show task vectors for clockwise (CW) trajectories, counterclockwise (CCW) trajectories, and all trajectories considered together, respectively. We observe that CW and CCW circle trajectories are represented on two manifolds in the 2D space.

In Fig. 10 in the paper, we show the geometry of task vectors for the Circular-Trajectory problem for sequences where $c = -1$, *i.e.*, the trajectories are clockwise (CW). In this section we look at

the geometry of task vectors for sequences with counterclockwise (CCW) trajectories, as well as all sequences considered together. In Fig. 34, we plot the 2D projection of task vectors for CW, CCW and both taken together, respectively. We consider 32 radius values while training. The variance explained by 2 PCs for the three plots is $93.55\%, 96.32\%, 89.1\%$, respectively. Darker colors correspond to smaller radius values, and the two colormaps in the third subplot represent CW or CCW trajectories. We observe that CW and CCW circle trajectories are represented on two manifolds in the 2D space.

**Training Details.** In this setting, we sample a new batch of sequences at every iteration. We use batch size 64 and train the model on a total of $200\,000$ sequences. We use Adam optimizer with learning rate $10^{-4}$ and weight decay 0.001. We use representations at position $\lfloor \frac{n}{2} \rfloor$ as the task vector following the process mentioned in Section 3.1, averaging over 100 sequences each. These settings remain the same for the experiments in the next section.

### B.3 MORE ON THE RECTANGULAR-TRAJECTORY PROBLEM

In this section we consider a Rectangular-Trajectory problem, parameterized by two parameters, namely the lengths of the two sides of the rectangle, say $(a, b)$. Specifically, the trajectories contain points on axis-aligned rectangles centered at the origin. Let $e$ denote the number of points on each edge of the rectangle spaced uniformly. The starting point of the sequence is randomly sampled from one of the $e$ points on the right vertical edge of the rectangle. The rest of the points are obtained by traversing the rectangle CW or CCW, determined by $c = -1$ or 1. Fig. 35 shows an example sequence.

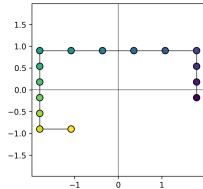

Figure 35: An illustration of a sequence of points for the Rectangular-Trajectory problem with $e = 5$ points per edge, and $n = 15$. See text for details.

Similar to Circular-Trajectory problem, each sequence is obtained by first sampling $a$ and $b$ uniformly between 1 and 4, then sampling the starting point and $c = -1$ or 1, and then following the aforementioned process. For our experiments, we set $e = 5$ and $n = 15$ for this task. The number of tasks $K$ denotes the number of different combinations $(a, b)$.

In Fig. 36, we plot the 2D projection of the task vectors obtained for all $(a, b)$ combinations lying on the 2D grid between $a \in [1, 4]$ and $b \in [1, 4]$. Similar to the experiments in Fig. 10, we plot the task vectors for trajectories with $c = -1$ here. We consider $K = 32$ in this experiment. The first two PCs explain $91.97\%$ variance. We observe that all the task vectors lie on a 2D manifold.

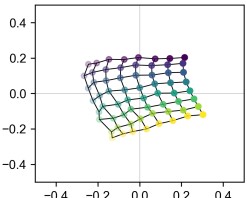

Figure 36: 2D projection of the task vectors obtained for 64 $(a, b)$ combinations. Fixed color or transparency level corresponds to fixed $a$ or $b$, respectively. The task vectors lie on a (smooth) 2D manifold.

This setting goes beyond the Circular-Trajectory problem and shows that transformers represent task vectors corresponding to the problem parameters (radius for circles and edge lengths for rectangles) in low-dimensional (smooth) manifolds in both cases.

## C    FURTHER DISCUSSIONS

In this section, we discuss potential implications of our discoveries on finer (theoretical) understanding of transformer models.

- **Multi-hop ICL**. Wang et al. (2024) found that small transformers trained from scratch on explicit instruction-based multi-hop problem have significant difficulty learning the bridge concepts and obtain high generalization accuracy. This is somewhat expected: the "hops" on their synthetic knowledge graph are close to *permutation* functions, and directly learning compositions of permutations is known to be hard. Yet, we show in our work that *LLMs in the wild* do utilize hidden bridge-concept embeddings effectively. A hypothesis to explain this is that, the ability to learn generalizing representations is *not* intrinsic to the transformer architecture, and gradient descent only discovers compressed representations which transcend instance-specific information under *certain data-structure and model-scale conditions*. To some extent, this also suggests that pure *expressive-power-type theoretical results* might yield limited conclusions on the generalization abilities of transformer models.

- **Continuous-parameterization ICL**. One of the most intriguing observations in this set of experiments is the *order-preserving* nature of the task vector manifold in the trained transformer models. Essentially, gradient descent was able to correctly compress the large number of tuples of inputs into sparse points which fall on a low-dimensional manifold, and these points follow the order relation of the integers. It could be interesting to theoretically and empirically understand how such representations arise during training, and why representation linearity appears to *break* as task complexity increases beyond a threshold (e.g., the line evolves into a *helix* in the offset ICL experiment).

## D    USE OF LARGE LANGUAGE MODELS (LLMS)

Frontier LLMs assisted in the process of curating the Country and Company datasets for our multi-hop ICL experiments discussed in Section 2 and in Appendix A above. As we discussed at the beginning of Appendix A.2, we rely on ChatGPT o3 (available at the time of dataset curation) to perform the initial sampling of the countries and companies' source and target entity data. Manual filtering and cleaning of the datasets were then performed.

We did not use LLMs for writing, ideation, or finding related work.

