# OpenReview forum: "Latent Concept Disentanglement in Transformer-based Language Models"
_ICLR.cc/2026/Conference — ICLR 2026 Poster_

### Official Review · Reviewer_e7J6 · 2025-10-31

**Soundness:** 3
**Presentation:** 3
**Contribution:** 3
**Rating:** 8
**Confidence:** 3

**Summary:**

This paper investigates how LLMs (specifically, Gemma-2-27B) represent and manipulate latent concepts when performing ICL. The authors combine mechanistic interpretability methods with controlled synthetic tasks to explore whether models infer and compose hidden concepts (e.g., countries, radii) in structured ways. The authors explore two major experimental setups: two-hop reasoning tasks and geometric tasks, and conclude that transformers naturally form disentangled, low-dimensional concept representations that are causally and geometrically interpretable.

**Strengths:**

* The division between discrete and continuous latent concepts is conceptually clean and helps generalize across types of reasoning.

* Identifying sparse, interpretable sub-circuits for latent concept composition is relevant to understanding compositional generalization.

* The authors situate their findings well within recent discussions of task vectors, linear representation hypothesis, and in-context reasoning circuits.

* Demonstrating low-dimensional manifolds for latent numerical parameters (and the ability to steer them) is a compelling visualization of representational structure.

**Weaknesses:**

* While synthetic setups isolate mechanisms effectively, they may oversimplify real-world linguistic reasoning. It’s unclear whether these disentanglement findings generalize to naturalistic contexts.

* The discrete reasoning experiments rely on Gemma-2 models, while continuous ones use small custom transformers. The gap in architecture and scale complicates unification. Moreover, would the discrete reasoning results replicate with other models?

* Activation patching is compelling, but causal claims might be overstated without broader ablation or counterfactual consistency checks.

* While qualitative visualizations (e.g., PCA plots) are persuasive, more statistical rigor (variance, replicability, robustness to seeds) would add credibility.

* The discussion of how these insights could enhance controllability, generalization, or interpretability in applied settings remains mostly speculative.

**Questions:**

* Do you observe similar disentanglement patterns in larger, more naturalistic multi-hop datasets (e.g., HotpotQA, factual chains beyond two hops)?

* Are the same attention heads consistently responsible for “bridge” resolution across tasks or prompts, or does specialization vary by context?

* How might identifying latent concept circuits inform real-world model editing, debiasing, or steering applications?

---

> ### Author Response · Authors · 2025-11-21
> **Response to Reviewer e7J6, Part 1**
>
> Thank you for your constructive feedback! We are encouraged that you appreciate our methodology and mechanistic insights, and acknowledge the research problem’s relevance and importance. Below are our responses to your questions and comments.
>
> > W1 & Q1: “While synthetic setups isolate mechanisms effectively, they may oversimplify real-world linguistic reasoning. It’s unclear whether these disentanglement findings generalize to naturalistic contexts.”, “Do you observe similar disentanglement patterns in larger, more naturalistic multi-hop datasets (e.g., HotpotQA, factual chains beyond two hops)?”
>
> Thank you for raising these interesting points regarding the limitations of and potential directions to extend our work, we have flagged them more clearly in the conclusion of our paper now.  We wish to kindly note that the tasks involved in this work are deliberately simplified representations of real-world use cases of LLMs. The goal of this paper is to isolate and precisely understand the _mechanisms_ by which transformers can _disentangle and manipulate latent concept representations during in-context learning (ICL)_. The controlled settings of our 2-hop reasoning and continuous-parameterization ICL tasks allow us to
>
> (i) keep the ground truth concepts _unambiguous_, and
>
> (ii) perform precise _interventions_ and analyze the _geometry_ of representations.
>
> Starting directly from naturalistic settings (e.g., open-ended multi-hop QA like HotpotQA) would introduce many _confounders_ and make it far harder to draw clean conclusions about whether the model is genuinely using a specific intermediate concept. Our more controlled approach is intended as a _foundational_ step: it provides a rigorous testbed for concept-level mechanisms, on top of which future work can build toward more realistic tasks. This staged strategy is standard in mechanistic interpretability.
>
> We also note that there is partial evidence of disentangled “concept” features in certain naturalistic contexts, where the two-hop relation is explicitly specified in the prompt. For example, [1] identified “Texas” features that strongly fire when Claude 3.5 Haiku is prompted with “Fact: the capital of the state containing Dallas is…”. Overall, understanding concept representations in prompts that implicitly or explicitly specify latent concepts is an active research direction; our work belongs to the implicit case, which is currently much less studied.
>
> >W2 (i): The discrete reasoning experiments rely on Gemma-2 models, while continuous ones use small custom transformers. The gap in architecture and scale complicates unification.
>
> As noted in our related work section (lines 183 - 187), there is a contemporaneous work [2] which studies the task vectors a Llama model uses for solving the add-$k$ problem. We provide complementary results for smaller models, where we have full control over training, and can hence conclude that the geometry of the task vectors actually arises from the latent task structure. Moreover, we offer deeper insights on the low-dimensional and order-preserving nature of the transformer’s task representations, in the circular- and rectangular-trajectory problems, with similarly controlled experiments for clean mechanistic insights.
>
> > W2 (ii) Would the discrete reasoning results replicate with other models?
>
> We expect the discrete multi-hop reasoning results to qualitatively replicate in other LLM families, although model-specific nuances likely exist. For example, some families may exhibit sparser and more localized components for resolving latent concepts than others. However, due to limited computational resources, we chose to focus on in-depth analysis of a single LLM family (Gemma-2). This allowed us to: (i) **consistently localize concept invocation and composition mechanisms** across two 2-hop ICL datasets (Geography and Company), and (ii) observe a clear **scaling trend in disentanglement with respect to model size** within one training pipeline.
>
> We believe our choice to prioritize depth and rigor within one family is reasonable and in line with prior mechanistic interpretability studies that focus on a single (family of) models in controlled settings [2-5].

---

> ### Author Response · Authors · 2025-11-21
> **Response to Reviewer e7J6, Part 2**
>
> > W3 & W4: “Activation patching is compelling, but causal claims might be overstated without broader ablation or counterfactual consistency checks.”, “While qualitative visualizations (e.g., PCA plots) are persuasive, more statistical rigor (variance, replicability, robustness to seeds) would add credibility.”
>
> We agree that causal claims should be backed by systematic, rather than purely anecdotal, evidence. Beyond the main-text examples, _we do provide extensive statistical analyses in the appendices_:
>
> - In Appendix A.2 (Geography) and A.4 (Company), we present _activation patching_ experiments that sweep over _all_ combinations of source and target types (Figures 12-26 and 28-30). We observe consistent evidence of _causal transferability_ of the concept embeddings and identify mediators consistently exhibiting strong indirect effects across these settings.
> - In Appendix A.3, we report inter- and intra-concept _correlation_ strengths on samples generated from the same versus different bridge concepts across _all_ source-target types. These quantitative results corroborate the qualitative disentanglement patterns shown in the main-text PCA visualizations.
> - Figures 31, 32, and 34 and the textual explanations in Appendix B expand on the numerical-concepts experiments, demonstrating the stability of the low-dimensional manifolds and steering behavior across seeds and parameter settings.
>
> We have updated the main text to make more explicit reference to these appendix results. Moreover, due to the controlled prompt format, well-defined source and target types, and unambiguous answers in our ICL instances, we have effectively minimized confounding variables between the (distribution of) prompt pairs which we perform activation patching on to localize the model components.
>
> > Q2: Are the same attention heads consistently responsible for “bridge” resolution across tasks or prompts, or does specialization vary by context?
>
> Based on our observations on the two datasets, we find that there is _consistency_ in the attention heads which resolve the latent concepts. As shown in Appendix A.2 and A.3, head groups (24,30;31) and (35,22;23) have high causal scores on _both_ datasets, measured with respect to the logit-difference metric and rank-steering metric.
>
> A subtler point is that, for the _Company_ dataset, a greater number of attention heads are required for strong intervention results (6 head groups instead of 2 as in the Geography case), showing that the latent concept encoding for companies appears more _diffuse_ than for countries. Another intriguing observation from Figure 30 in Appendix A.3 is that, head group (35,22;23) exhibits weak disentanglement on [Product, Founder] problems (while maintaining good disentanglement strength on other types), indicating that the company concepts are _not_ encoded in as “universal” a manner across source-target types as the country concepts, showing weaker robustness of the representation.
>
> The more _diffuse_ encoding of the latent concept, coupled with somewhat _weaker_ disentanglement, offer a plausible explanation for Gemma-2-27B’s _lower accuracies_ on the Company dataset: the [Product, Founder], [Product, HQ City] and [Founder, HQ City] problems have accuracies 73.5%, 80.7%, 88.6% respectively, significantly lower than the accuracies on the Geography puzzles, which mostly concentrate around 90% or above.

---

> ### Author Response · Authors · 2025-11-21
> **Response to Reviewer e7J6, Part 3**
>
> > Q3 & W5: “How might identifying latent concept circuits inform real-world model editing, debiasing, or steering applications?”, “The discussion of how these insights could enhance controllability, generalization, or interpretability in applied settings remains mostly speculative.”
>
> We agree this is an important question. Our work is primarily a _foundational_ study: it aims to provide rigorous evidence that, in ICL, LLMs rely on _modular_, _reusable_ components and embedding directions that mediate _hidden concepts_ in their text outputs. While results from controlled settings do not automatically generalize to all real-world applications, we see two concrete ways in which concept-level circuits can inform practice:
> 1. **Representation-level monitoring**. Text-level explanations (chain-of-thought) can be unfaithful [6-8]. Concept-level circuits and task vectors offer a complementary tool: they allow us to inspect which latent concepts are actually used during reasoning, detect when a model starts relying on undesired/spurious concepts, and potentially design monitors that trigger when specific concept directions become active. This could be especially useful in example-based ICL, since the LLM may well mis-interpret what latent concepts to use from the examples.
> 2. **Concept-level steering**. Once a circuit mediating a particular concept is identified, we can, in principle, intervene on that circuit or its associated directions. Our results in controlled ICL settings show that, when using concept embeddings extracted from prompts with sufficiently many demonstrations, the interventions can systematically alter certain hidden concepts in the predictions, which is a necessary precondition for real-world editing and debiasing at the _representation_ level.
>
> **References**.
> 1. J. Lindsey et al. On the Biology of a Large Language Model. Transformer Circuits, 2025.
>
> 2. X. Hu et al. Understanding In-context Learning of Addition via Activation Subspaces. ArXiv Preprint, 2025.
>
> 3. K. Wang et al. Interpretability in the Wild: a Circuit for Indirect Object Identification in GPT-2 small. ICLR 2023.
>
> 4. A. Stolfo et al. A Mechanistic Interpretation of Arithmetic Reasoning in Language Models using Causal Mediation Analysis. EMNLP 2023.
>
> 5. M. Lu et al. Paths Not Taken: Understanding and Mending the Multilingual Factual Recall Pipeline. ArXiv Preprint, 2025.
>
> 6. M. Turpins et al. Language Models Don’t Always Say What They Think: Unfaithful Explanations in Chain-of-Thought Prompting. NeurIPS 2023.
>
> 7. Y. Chen et al. Reasoning Models Don’t Always Say What They Think. ArXiv Preprint, 2025.
>
> 8. T. Korbak et al. Chain of Thought Monitorability: A New and Fragile Opportunity for AI Safety. ArXiv Preprint, 2025.

---

### Official Review · Reviewer_9BHQ · 2025-10-31

**Soundness:** 3
**Presentation:** 3
**Contribution:** 3
**Rating:** 6
**Confidence:** 3

**Summary:**

This paper studies the problem of whether and how transformers represent latent concepts as part of their computation in a an ICL setup.

Key ideas

* Use activation patching (a.k.a Causal Mediation Analysis) to determine if the model is indeed representing the latent concept in a discrete multi-hop setting.
* For the numerical latent variables, the authors propose the use of linear probing the intermediate model embeddings to assert the existence of task vectors.
* Use PCA and project the task vectors onto the first two principle components and then study the geometry of the resultant manifolds along with the ordering of the points. Using steering (or interpolation) based methods to show that model captures concept’s geometry.

Main contributions

* Use of causal and correlation techniques to show that for tasks requiring memorized world knowledge, a sparse set of attention heads are responsible for resolving intermediate latent concept, thereby rejecting the hypothesis that the model takes a shortcut (directly maps the input to the corresponding output without any intermediate steps). This shows that the model is indeed performing abstract reasoning (chained reasoning) in its hidden activations.
* Show how the above phenomenon plays out in case of smaller language models are used (2B, instead of 27B) and larger number of ICL examples.
* In alignment with the previous work on Linear Representation Hypothesis (LRH), the authors establish the existence of task vectors. In addition to that, the authors show that these representations reflect the geometry of the latent variable.

**Strengths:**

Originality

* The unorthodox evaluation methodology of using activation patching with counterfeit examples is creative.
*  The technique of using interpolation (referred in paper as steering) in this setting is innovative and establishes the geometry of the underlying latent variable.
* The paper discusses the findings of previous studies in the literature survey section and clearly state the novel contributions of this work.

Quality

* Overall, I find the paper to be of good quality, the methodology is sound and the results are well presented. The experiments are conducted on a variety of datasets which demonstrates the robustness of the claims.

Clarity

* The paper is well written and easy to read. The ideas are explained clearly with supporting experiments.

Significance

* This works brings us closer to understanding how LLMs represent intermediate concepts, perform chained reasoning in ICL setup.
* It uses tasks such as add-k, circular trajectory, rectangular trajectory to empirically prove that for numerical latent variable problems, tasks vectors indeed represent the geometry of latent variables.
* The paper strengthens the broader hypothesis that LLMs engage in multi-step reasoning rather than relying solely on direct input-output associations.
* The authors also highlight the influence of model size and the number of ICL examples on the latent structures.

**Weaknesses:**

* In the “memorized world knowledge” task, the authors claim that the LLM relies on step-by-step composition of latent concept. However, all their experiments are of the kind where the latent concept has a one-to-one mapping with the output class (for instance country → capital). While, there is no problem with this choice, it might be helpful to study different scenarios (such as country (bridge) → famous politicians from that country)
* In Figure 10, for the circular trajectory problem, even with high value of beta (0.8), we find that MSE corresponding to original and opposite radius is nearly equal. Isn’t this counter-intuitive?

**Questions:**

* In the “Memorized world knowledge” thread - While the existing work uses 2-hop reasoning tasks, it would be interesting to study higher order transitive tasks (3 hops or more). It would be interesting to tease out the impact of model size  on such tasks. One potentially interesting finding could be that LLMs are able to deal with k-hops (k>>2) reasoning tasks much easily as compared to small language models (SLMs). It would be interesting to see, if LLMs indeed contain sparse attention heads which represent all the intermediate (bridge) concepts.
* The authors have used two variants of Gemma 2 model family - 27B and 2B. It would be helpful to conduct experiments using models of varying capacity - including a few with multilingual support, so that claims/findings can be strengthened further.
* In section 3, add-k problem, I did not understand the reason behind the constraint k_{i+1} - k_i = 3. What happens when we relax the constraint? Will the claims be still valid, if this constraint is not met?
* The authors mention that ideal scaling constant for bridge intervention is 2.0. My understanding is scaling is a way to boost the importance of the signal (bridge information). Is that correct? If that is so, it is understandable that the performance saturates at 4.0. But, what could be the reason for deterioration?

---

> ### Author Response · Authors · 2025-11-21
> **Response to Reviewer 9BHQ, Part 1**
>
> Thank you for your constructive feedback! We are encouraged that you find our paper well-written, appreciate our methodology and mechanistic insights, and acknowledge the research problem’s importance. Below are our responses to your questions and comments.
>
> > W1 & Q1 & Q2: “In the ‘Memorized world knowledge’ thread - While the existing work uses 2-hop reasoning tasks, it would be interesting to study higher order transitive tasks (3 hops or more) … It would be interesting to see, if LLMs indeed contain sparse attention heads which represent all the intermediate (bridge) concepts”, “… It might be helpful to study different scenarios (such as country (bridge) → famous politicians from that country)”, “… It would be helpful to conduct experiments using models of varying capacity - including a few with multilingual support, so that claims/findings can be strengthened further.”
>
> Thank you for these interesting directions to extend our work. We agree that higher-order transitive tasks, bridge concepts with many-to-many mappings, and a broader set of model families are promising directions. Our current choice to focus on 2-hop (instead of "3+"-hop) reasoning with unique outputs in English is mainly driven by computational constraints. Even in this setup, providing rigorous causal and correlational evidence for disentangled hidden concepts requires extensive activation-patching sweeps over many source-target combinations and model components. For the Geography and Company concepts, the full results are shown in Figures 1–4, 13–26, and 28–30.
>
> For $k$-hop tasks with $k > 2$ and/or ambiguous outputs, we would need to sweep over many more source-target combinations for each hidden reasoning step and, for each component and sample, assess how interventions change _sets_ of candidate answers in the logits (if the answers are non-unique). This quickly becomes much more expensive, and if the outputs are not unique, assessing intervention effects would likely require LLM or human judges, which is beyond our current resources.
>
> We therefore prioritized what we see as an important first step: _carefully understanding how transformers utilize hidden concepts in a clean 2-hop ICL setting_. This allowed us to (i) _**consistently localize** concept invocation and composition mechanisms_ across two 2-hop ICL datasets (Geography and Company) within one LLM family, and (ii) observe a _clear **scaling trend** of disentanglement with respect to model size_.
>
> Finally, we note that focusing on a single (family of) LLM in highly controlled settings is common in mechanistic interpretability for prioritizing rigor and depth over breadth of analysis [1-4].
>
> > W2: In Figure 10, for the circular trajectory problem, even with high value of beta (0.8), we find that MSE corresponding to original and opposite radius is nearly equal. Isn’t this counter-intuitive?
>
> In Fig. 10, we report the MSE with respect to the original, the target and the ‘opposite’ radius values. The key observation is that for any $\beta$, the MSE with respect to the target radius is the smallest among the three. As $\beta$ increases, the target radius gets closer to the ‘opposite’ radius. Therefore, the MSE with respect to the original radius increases while the MSE with respect to the ‘opposite’ radius drops. It is also important to note that in the circular trajectory setting, the task vectors do not lie in a straight line, in contrast to the add-k setting, where the task vectors lie almost in a line and linearly encode different offset/task values. This geometric difference could explain why the steering coefficients in this setting behave differently, and a larger $\beta$ value is required to steer the model towards the ‘opposite’ radius.
>
> > Q3: In section 3, add-k problem, I did not understand the reason behind the constraint k_{i+1} - k_i = 3. What happens when we relax the constraint? Will the claims be still valid, if this constraint is not met?
>
> For the add-$k$ problem, we consider the specific setting $k_1=1$ and $k_{i+1}-k_i=3$ purely as a convenient choice to obtain well-separated and evenly spaced offset values. Our findings in this setting, that the task vectors encoding offset information lie on a nearly 1-dimensional manifold, rely on the fact that the tasks are parameterized by scalar offsets with an underlying linear relation, not on the specific spacing between consecutive $k_i$.

---

> > ### Comment · Reviewer_9BHQ · 2025-11-24
> >
> > Thank you for your response. Your response satisfactorily addresses my comments and questions.

---

> ### Author Response · Authors · 2025-11-21
> **Response to Reviewer 9BHQ, Part 2**
>
> > Q4: The authors mention that the ideal scaling constant for bridge intervention is 2.0. My understanding is scaling is a way to boost the importance of the signal (bridge information). Is that correct? If that is so, it is understandable that the performance saturates at 4.0. But, what could be the reason for deterioration?
>
> Thank you for this intriguing question! This phenomenon suggests that the causal efficacy of a concept embedding in a transformer has certain nonlinear effects on the downstream computations, and can cause errors instead of boosting a certain concept cleanly. We see at least three (not mutually exclusive) hypotheses:
> 1. **Imperfect disentanglement**. Disentanglement is imperfect in these models, so as the scaling constant goes beyond a certain point, the elevated _noise or spurious signals_ can cause the model to shift its answer towards certain problematic directions. The imperfect disentanglement can be observed in our correlation results, and attributed to the fact that there likely is a _scaling_ trend: the 2B model exhibits much weaker and noisier disentanglement with respect to the bridge concepts in both the causal-intervention and correlation-based experiments, so it is natural to suspect that the 27B model is along the scaling trajectory of increasingly better but still _imperfect_ disentanglement of hidden concepts.
> 2. **Normal scaling only in $\epsilon$-neighborhoods**. It is possible that scaling certain concept directions too strongly moves the resulting activation off the natural manifold too much, causing the downstream computations to behave in unexpected manners. This is possible if the linear representation hypothesis only holds in a certain $\epsilon$-neighborhood of (certain regions of) the natural activation manifold.
> 3. **Inherently nonlinear manifolds**. The activation manifold might be inherently nonlinear. More specifically, in the small transformer experiments, we noticed that as the add-$k$ task grows in the complexity (the number of concepts the model needs to learn, parameterized by the number of offsets), the learnt manifold is no longer a simple line but a _helix_ (Appendix B.1). Linearly interpolating or scaling points on this helix would travel “off-manifold”, and would _not_ be expected to produce “smooth” outputs when the scaling constants become too large. This suggests that valid linear interpolation/scaling of the latent concept representations might _break_ down as the complexity or number of latent concepts increases and scaling constant (during intervention) goes up, assuming that the model capacity is held constant.
>
> **References**.
>
> 1. K. Wang et al. Interpretability in the Wild: a Circuit for Indirect Object Identification in GPT-2 small. ICLR 2023.
>
> 2. A. Stolfo et al. A Mechanistic Interpretation of Arithmetic Reasoning in Language Models using Causal Mediation Analysis. EMNLP 2023.
> 3. M. Lu et al. Paths Not Taken: Understanding and Mending the Multilingual Factual Recall Pipeline. ArXiv Preprint, 2025.
> 4. X. Hu et al. Understanding In-context Learning of Addition via Activation Subspaces. ArXiv Preprint, 2025.

---

> > ### Comment · Reviewer_9BHQ · 2025-11-24
> >
> > Thank you for your response. Maybe for the future - It would be interesting to see which of these three hypothesis is the reason for the deterioration.

---

### Official Review · Reviewer_4uha · 2025-10-31

**Soundness:** 3
**Presentation:** 3
**Contribution:** 3
**Rating:** 6
**Confidence:** 3

**Summary:**

This paper systematically studies how large language models represent, disentangle, and utilize latent concepts in in-context learning. The authors use a series of controlled tasks combined with mechanistic interpretability methods to provide both causal and correlational evidence, revealing the internal reasoning and representation mechanisms of Transformer models when solving ICL tasks.

**Strengths:**

The exploration of how LLMs understand and reason at the conceptual level is interesting and provides valuable insights for the community. The use of causal mediation analysis and PCA visualization to validate the claims is persuasive under the synthetic tasks presented in the paper. The experimental design is rigorous and highly interpretable. In particular, the demonstration of low-dimensional manifold structures in the mode’s representation space (i.e., the geometric interpretability of latent variables) is a novel perspective.

**Weaknesses:**

The tasks and experimental setups are overly idealized, relying almost entirely on highly synthetic toy tasks, which do not represent real-world natural language reasoning tasks. Of course, this is a common issue in the interpretability field.

The analysis of model scale and generalization is insufficient. The paper only compares Gemma-2-27B and 2B, without systematically examining whether the same mechanisms hold across different architectures (e.g., LLaMA, Qwen series) or larger-scale models. It is also unclear why the experiments in Section 3 are conducted on a small GPT-2–style model rather than the previously used Gemma models, and what the motivation for this model substitution is.

The paper provides only empirical analysis. It would be more compelling if any theoretical or mechanistic explanation were offered for why Transformers naturally form linear geometric task vectors or bridge concepts.

Are the bridge concepts unique? The experimental design in the paper is overly simple, but in real-world scenarios, the relationships between concepts are much more complex. With only a few in-context examples, there may exist multiple potential bridge concepts between the source and target, a possibility that the paper does not discuss.

Another concern is how such concept-level empirical interpretability analyses can provide meaningful value for real-world applications. In other words, when would we actually need concept-level explanations? Even if we know which attention heads encode certain concept representations, how should we intervene or utilize them in practice? Could specific application scenarios be provided as examples?

**Questions:**

Please see Weaknesses.

---

> ### Author Response · Authors · 2025-11-21
> **Response to Reviewer 4uha, Part 1**
>
> Thank you for your valuable feedback! We are glad that you appreciate the paper’s problem setting, methodology and mechanistic insights on the representations of transformer models. We will address your comments below.
>
> > W1: The tasks and experimental setups are overly idealized, relying almost entirely on highly synthetic toy tasks, which do not represent real-world natural language reasoning tasks. Of course, this is a common issue in the interpretability field.
>
> Thank you for this point, we now flag this more clearly as a limitation in the revised paper's conclusion section. We wish to kindly note that our goal in this work is to isolate and precisely understand the mechanisms by which transformers can disentangle and manipulate latent concept representations during in-context learning (ICL). The controlled 2-hop reasoning and continuous-parameterization ICL tasks allow us to (i) keep the ground-truth latent concepts _unambiguous_, and (ii) perform _precise causal interventions_ and _geometric analyses of the learned representations_.
>
> Directly initiating the study in an uncontrolled, open-ended setting would introduce numerous _confounding factors_, making it much harder to draw clean conclusions about whether the model is truly using the intended intermediate concepts. We therefore view our synthetic setups as a _foundational_ step: they provide a rigorous testbed for concept-level mechanisms, on top of which future work can build toward more naturalistic language tasks. This staged approach is standard in mechanistic interpretability.
>
> > W2 (i): The analysis of model scale and generalization is insufficient. The paper only compares Gemma-2-27B and 2B, without systematically examining whether the same mechanisms hold across different architectures (e.g., LLaMA, Qwen series) or larger-scale models
>
> Due to the constraints in our computing resources, we focused on conducting in-depth, rigorous analysis on one LLM family, instead of shallow analysis on a number of LLM families. This deeper analysis shows:
>
> 1. _**Consistent localization** of concept invocation and composition mechanisms_ on two 2-hop ICL datasets (Geography and Company) within Gemma-2-27B, with _exhaustive_ evidence over all source-target combinations shown in Figures 1 - 4, 13 - 26, and 28 - 30. This is already computationally demanding, due to the large number of patching sweeps required;
> 2. A _**scaling trend** of disentanglement with respect to model size_, comparing Gemma-2-27B against Gemma-2-2B (Appendix A.2, Figure 27), finding that the _smaller_ LLM variant contains a much _weaker_ and _noisier_ version of the bridge-resolving mechanism. Comparing models within one family, which share training data and pipeline, gives a clean “same architecture but scaled” comparison.
>
> We believe our choice to prioritize depth and rigor within one family is reasonable and in line with prior mechanistic interpretability studies that focus on a single (family of) models in controlled settings [1-5].
>
> > W2 (ii): It is also unclear why the experiments in Section 3 are conducted on a small GPT-2–style model rather than the previously used Gemma models, and what the motivation for this model substitution is.
>
> As noted in our related work section (lines 183 - 187), there is a contemporaneous work [4] which studies the task vectors a Llama model uses for solving the add-k problem. We provide complementary results for smaller models, where we have full control over training, and can hence conclude that the geometry of the task vectors only arises from the latent task structure. Moreover, we offer deeper insights on the _low-dimensional_ and _order-preserving_ nature of the transformer’s task representations, in the _circular- and rectangular-trajectory_ problems, with similarly controlled experiments for clean mechanistic insights.

---

> ### Author Response · Authors · 2025-11-21
> **Response to Reviewer 4uha, Part 2**
>
> > W3: The paper provides only empirical analysis. It would be more compelling if any theoretical or mechanistic explanation were offered for why Transformers naturally form linear geometric task vectors or bridge concepts.
>
> Thank you for this point. We agree that understanding _why_ linear task vectors and bridge concept representations arise is a central open problem. Providing a full theoretical account of this phenomenon across large LLMs is beyond the scope of our current work, which is focused on establishing rigorous empirical evidence that such structures exist and are causally used in real models and tasks.
>
> That said, our experimental results do have implications on further theoretical and empirical analyses of transformers.
>
> 1. **Multi-hop ICL**. [5] found that small _transformers trained from scratch_ on explicit instruction-based multi-hop problem have significant _difficulty_ learning the bridge entities and obtain high generalization accuracy. This is somewhat expected: the “hops” on their synthetic knowledge graph are close to permutation functions, and directly learning compositions of permutations is known to be hard. Yet, we show in our work that _LLMs in the wild_ do utilize hidden bridge-concept embeddings effectively. This suggests that the _ability to learn generalizing representations is not intrinsic to the transformer architecture_, and gradient descent only discovers compressed “concept” representations under _certain data-structure and model-scale conditions_. This also shows that pure _expressive-power-type theoretical results_ likely yields limited conclusions on the generalization abilities of transformer models.
> 2. **Continuous-parameterization ICL**. One intriguing observation in this set of experiments is the _order-preserving_ nature of the task representation manifold in the trained transformer models. Essentially, gradient descent was able to correctly compress the large number of tuples of inputs into sparse points in the embedding space which fall on a _low-dimensional smooth manifold_, and these points follow the _order relation of the integers_. It could be interesting to (theoretically) analyze how such representations arise during training, and why the linear representations appear to break as task complexity increases beyond a threshold (e.g., the line evolves into a helix in the offset ICL experiment, shown in Appendix B.1).
>
> We added a discussion articulating such connections more explicitly in the revised manuscript’s Appendix C. We hope this will be a useful stepping stone for future analytical works in this direction.
>
> > W4: Are the bridge concepts unique? The experimental design in the paper is overly simple, but in real-world scenarios, the relationships between concepts are much more complex. With only a few in-context examples, there may exist multiple potential bridge concepts between the source and target, a possibility that the paper does not discuss.
>
> Thank you for raising this point. We acknowledge that it is a limitation of this work: we primarily worked with concepts which are unique, and focused our study in the regime with sufficient in-context samples such that the latent concept is unambiguous. We hypothesize that some form of “superposition” of disentangled features arises. We also suspect that this is a regime where disentanglement might fail more easily, due to the more challenging nature of the problems. These are important future directions to study.

---

> ### Author Response · Authors · 2025-11-21
> **Response to Reviewer 4uha, Part 3**
>
> > W5: Another concern is how such concept-level empirical interpretability analyses can provide meaningful value for real-world applications. In other words, when would we actually need concept-level explanations? Even if we know which attention heads encode certain concept representations, how should we intervene or utilize them in practice? Could specific application scenarios be provided as examples?
>
> We appreciate this concern and agree that practical uses of concept-level interpretability are still at an early stage. Our work is primarily a _foundational_ study: it aims to provide rigorous evidence that, in ICL, LLMs rely on _modular, reusable_ components and embedding directions that _mediate hidden concepts in their text outputs_. We believe that the ability to interpret, monitor, and steer transformers at the representation level will be increasingly important, especially as LLMs are deployed in higher-stakes settings. While results from controlled settings do not automatically generalize to all real-world applications, we see two concrete ways in which concept-level circuits can already inform practice:
>
> **Representation-level monitoring**. Text-level explanations (chain-of-thought) can be unfaithful [6-8]. Concept-level circuits and task vectors offer a complementary tool: they allow us to inspect which latent concepts are actually used during reasoning, detect when a model starts relying on undesired/spurious concepts, and potentially design monitors that trigger when specific concept directions become active. This is particularly relevant for example-based ICL, where the model may misinterpret which latent concepts it should extract from the demonstrations.
>
> **Concept-level steering**. Once a circuit mediating a particular concept is identified, we can, in principle, intervene on that circuit (or its associated representations). Our results in controlled ICL settings show that, when using concept embeddings extracted from prompts with sufficiently many demonstrations, the interventions can systematically alter latent concepts in the predictions. This is a proof-of-feasibility step toward representation-level editing in more realistic applications.
>
> **References**.
>
> 1. K. Wang et al. Interpretability in the Wild: a Circuit for Indirect Object Identification in GPT-2 small. ICLR 2023.
>
> 2. A. Stolfo et al. A Mechanistic Interpretation of Arithmetic Reasoning in Language Models using Causal Mediation Analysis. EMNLP 2023.
>
> 3. M. Lu et al. Paths Not Taken: Understanding and Mending the Multilingual Factual Recall Pipeline. ArXiv Preprint, 2025.
>
> 4. X. Hu et al. Understanding In-context Learning of Addition via Activation Subspaces. ArXiv Preprint, 2025.
>
> 5. B. Wang et al. Grokked Transformers are Implicit Reasoners: A Mechanistic Journey to the Edge of Generalization. NeurIPS 2024.
>
> 6. M. Turpins et al. Language Models Don’t Always Say What They Think: Unfaithful Explanations in Chain-of-Thought Prompting. NeurIPS 2023.
>
> 7. Y. Chen et al. Reasoning Models Don’t Always Say What They Think. ArXiv Preprint, 2025.
>
> 8. T. Korbak et al. Chain of Thought Monitorability: A New and Fragile Opportunity for AI Safety. ArXiv Preprint, 2025.

---

### Official Review · Reviewer_qgy5 · 2025-11-01

**Soundness:** 3
**Presentation:** 3
**Contribution:** 3
**Rating:** 6
**Confidence:** 4

**Summary:**

The paper analyzes how LLMs represent and combine latent "concepts" during ICL. To analyze these interactions, they employ causal mediation analysis to identify which components of the model’s computation mediate the effect of particular input concepts on final predictions.

**Strengths:**

- The paper takes a systematic and well-executed approach to analyzing internal mechanisms of ICL.
- The approach offers interpretable, fine-grained insight into how contextual information propagates, complementing existing representation-based probing techniques.
- The study spans synthetic geometric reasoning tasks and natural language problems, demonstrating that the framework generalizes across distinct domains of conceptual structure.
- The visual analyses (mediation heatmaps and intervention results) are well presented and help illustrate which attention paths encode specific concepts.
- The work contributes methodologically by adapting tools from causal inference to the study of neural mechanisms in LLMs.

**Weaknesses:**

- The paper treats latent concepts as interpretable dimensions or attention patterns but never defines them formally.
- The causal model in the introduction ($F = R \circ C$) is not linked to the implemented CMA pipeline. $R$ and $C$ are just introduced without definitions. There is no derivation showing that the empirical mediation quantities estimated from activations correspond to components of this decomposition.
- Experiments are mostly conducted on synthetic or simplified reasoning datasets (two-hop relations at most).
  These provide clarity but limit conclusions about behavior on realistic compositional or linguistic tasks.
- The paper occasionally presents correlations in mediation strength as evidence of conceptual disentanglement. In my opinion, stronger validation, like causal interventions or counterfactual ablation, would be needed to corroborate these claims.
- No stability analysis or statistical testing is reported. It is unclear how consistent the discovered mediators are across runs or models.

**Questions:**

1. How are concept variables defined or selected for mediation?
2. How stable are the identified mediating paths across random seeds or different model checkpoints?
3. Intuitively, can the mediators identified via CMA be related to known attention-head clusters or activation subspaces discovered in prior ICL localization work?

---

> ### Author Response · Authors · 2025-11-21
> **Response to Reviewer qgy5, Part 1**
>
> Thank you for your thoughtful feedback! We are glad that you find our analysis approach systematic and well-executed, and appreciate our mechanistic insights. We will address your comments below.
>
> > W1 & W2: “The causal model in the introduction ($F = R \circ C$) is not linked to the implemented CMA pipeline. R and C are just introduced without definitions. There is no derivation showing that the empirical mediation quantities estimated from activations correspond to components of this decomposition.”, “The paper treats latent concepts as interpretable dimensions or attention patterns but never defines them formally.”
>
> Thank you for raising this point. We added clarifications on how the hypothesized model $R \circ C$ relates to the causal mediation analysis in Sections 2 and 3. The finer technical details, including how we interpret the CMA results are presented in Appendix A and B. Since the general reader of our paper is not necessarily familiar with causal mediation analysis and other tools in mechanistic interpretability, in the main text, we choose to present results in a more _intuitive_ manner, and focus more on discussing how the multi-hop hypothesis suggests the CMA analyses, and how we reject the single-hop hypothesis with our casual and correlational results. We highlighted the changes in the main text in blue for your convenience.
>
>
> > W3: Experiments are mostly conducted on synthetic or simplified reasoning datasets (two-hop relations at most). These provide clarity but limit conclusions about behavior on realistic compositional or linguistic tasks.
>
> Due to the constraints in our computing resources, we focused on conducting in-depth, rigorous analysis on one LLM family, instead of shallow analysis on a number of LLM families. This deeper analysis shows:
>
> (1) _**Consistent localization** of concept invocation and composition mechanisms_ on two 2-hop ICL datasets (Geography and Company) within Gemma-2-27B, with _exhaustive_ evidence over all source-target combinations shown in Figures 1 - 4, 13 - 26, and 28 - 30. This is already computationally demanding, due to the large number of patching sweeps required;
>
> (2) A _**scaling trend** of disentanglement with respect to model size_, comparing Gemma-2-27B against Gemma-2-2B (Appendix A.2, Figure 27), finding that the _smaller_ LLM variant contains a much _weaker_ and _noisier_ version of the bridge-resolving mechanism. Comparing models within one family, which share training data and pipeline, gives a clean “same architecture but scaled” comparison.
>
> We believe our choice to prioritize depth and rigor within one family is reasonable and in line with prior mechanistic interpretability studies that focus on a single (family of) models in controlled settings [1-4].
>
> > W4: The paper occasionally presents correlations in mediation strength as evidence of conceptual disentanglement. In my opinion, stronger validation, like causal interventions or counterfactual ablation, would be needed to corroborate these claims.
>
> We agree that results based on causal interventions (activation patching) are stronger, since we essentially want to show the “causal transferability” of these “concept embeddings” across problem instances and source and target types. _The majority of our experiments are indeed relying on causal intervention results_: the activation patching analyses over all the source-target type combinations on our datasets are presented in Appendix A.2 and A.4, in Figures 12 to 21 and 28 - 29.
>
> In addition, we do believe that the correlation results (cosine similarity heatmaps) complement the intervention results, since they help us visualize how well the features actually align in embedding space, in addition to their causal transferability. This also helps us link the “concept embeddings” found in our ICL setting to the _linear representation hypothesis_, since we found the identified representations to cluster strongly with respect to the interpretable (discrete/continuous) concepts.
>
> > W5 (i): No stability analysis or statistical testing is reported.
>
> We do provide statistical tests for the ICL experiments in the Appendix. In Appendix A.2,we present patching experiments sweeping through all the combinations of source and target types on the Geography dataset (spanning Figures 12 to 23 and 27), and A.4 for the Company dataset (Figures 27 - 30). We observe consistent evidence of causal transferability of the concept embeddings and mediators exhibiting strong indirect effects across these settings. In addition, in Appendix A.3, we show the inter- and intra-concept correlation strengths on samples generated from the same versus distinct bridge concepts, again across all source-target types. The observations visualized in the main text are corroborated by these results. Moreover, Figures 31, 32 and 34 in Appendix B expand on the numerical-concepts experiments. We added more explicit references to these results in the updated version of our paper.

---

> ### Author Response · Authors · 2025-11-21
> **Response to Reviewer qgy5, Part 2**
>
> > W5 (ii) “It is unclear how consistent the discovered mediators are across runs or models”, “How stable are the identified mediating paths across random seeds or different model checkpoints?”
>
> Based on our observations on the two datasets, we find that there is _consistency_ in the attention heads which resolve the latent concepts. As shown in Appendix A.2 and A.3, head groups (24,30;31) and (35,22;23) have high causal scores on _both_ datasets, measured with respect to the logit-difference metric and rank-steering metric, across all source-target-type combinations.
>
> A subtler point is that, for the _Company_ dataset, a greater number of attention heads are required for strong intervention results (6 head groups instead of 2 as in the Geography case), showing that the latent concept encoding for companies appears more _diffuse_ than for countries. Another intriguing observation from Figure 30 in Appendix A.3 is that, head group (35,22;23) exhibits weak disentanglement on [Product, Founder] problems (while maintaining good disentanglement strength on other types), indicating that the company concepts are _not_ encoded in as “universal” a manner across source-target types as the country concepts, showing weaker robustness of the representation.
>
> The more _diffuse_ encoding of the latent concept, coupled with somewhat _weaker disentanglement_, offer a plausible explanation for Gemma-2-27B’s _lower accuracies_ on the Company dataset: the [Product, Founder], [Product, HQ City] and [Founder, HQ City] problems have accuracies 73.5%, 80.7%, 88.6% respectively, significantly lower than the accuracies on the Geography puzzles, which mostly concentrate around 90% or above.
>
>
> > Q1 How are concept variables defined or selected for mediation?
>
> We primarily work with human-interpretable concepts in this work. As discussed in Section 1, we design two sets of experiments, one where the concepts involved require “world knowledge” (the multi-hop ICL setup), and one which relies more on elementary logic (the numerical ICL setup). Furthermore, we wished to understand how transformers handle latent discrete versus continuous concept variables during ICL, which also contributed to defining our choices of ICL tasks.
>
> After defining the tasks, for the LLMs, we conduct activation patching sweeps over the model components (attention heads and MLPs) to look for ones with the strongest indirect effects, as measured via logit differences and ranks pre- and post-intervention. We then interpret the intervention results, and discover that the transformers indeed capture the interpretable concepts in our data settings, as reflected in our analysis process in Sections 2 and 3, and Appendices A and B.
>
> > Q3 Intuitively, can the mediators identified via CMA be related to known attention-head clusters or activation subspaces discovered in prior ICL localization work?
>
> Yes. The surprising discovery is that in ICL, the transformers rely on certain hidden representations which transcend text-level instance-specific information to produce its text-level generations, and these representations cluster with respect to human-interpretable concepts. This strongly mirrors the types of concept directions found in other works in mechanistic interpretability which corroborate the linear representation hypothesis (such as truthfulness [5], refusal [6], geographic entities [7]). It is an interesting future direction to systematically analyze _how causally transferable and directionally (mis-)aligned the ICL concept directions are with the ones obtained on prompts which more explicitly specify the concepts_.
>
> **References**.
>
> 1. K. Wang et al. Interpretability in the Wild: a Circuit for Indirect Object Identification in GPT-2 small. ICLR 2023.
>
> 2. A. Stolfo et al. A Mechanistic Interpretation of Arithmetic Reasoning in Language Models using Causal Mediation Analysis. EMNLP 2023.
>
> 3. M. Lu et al. Paths Not Taken: Understanding and Mending the Multilingual Factual Recall Pipeline. ArXiv Preprint, 2025.
>
> 4. X. Hu et al. Understanding In-context Learning of Addition via Activation Subspaces. ArXiv Preprint, 2025.
>
> 5. S. Marks et al. The Geometry of Truth: Emergent Linear Structure in Large Language Model Representations of True/False Datasets. COLM 2024.
>
> 6. A. Arditi et al. Refusal in Language Models Is Mediated by a Single Direction. NeurIPS 2024.
>
> 7. J. Lindsey et al. On the Biology of a Large Language Model. Transformer Circuits, 2025.

---

> > ### Comment · Reviewer_qgy5 · 2025-11-26
> >
> > Thank you for the clarifications and references to the appendix material.
> >
> > My main technical concerns are all adequately addressed.
> >
> > I still view the reliance on controlled, synthetic tasks and a single LLM family as a limitation for direct generalization to more naturalistic settings. That said, I understand the substantial computational cost of the type of causal and mechanistic analyses performed here. For a foundational, mechanism-first study, the experimental scope is reasonable.
> >
> > Overall, the rebuttal and clarifications strengthen my confidence in the soundness and usefulness of the paper and address my main technical reservations. I have updated my score accordingly.

---

### Author Response · Authors · 2025-11-21
**General Response**

We wish to thank all four reviewers for their insightful feedback and helpful suggestions.

We are encouraged that they describe our analysis of in-context learning (ICL) as “systematic and well-executed” and “contribut[ing] methodologically by adapting tools from causal inference to the study of neural mechanisms in LLMs” (reviewer **qgy5**); that the demonstration of low-dimensional manifold structures in the model’s representation space is seen as “a novel perspective” (reviewer **4uha**); that the paper is well written, the “unorthodox evaluation methodology using activation patching … is creative,” and the insights on “the influence of model size and the number of ICL examples on the latent structures” are valuable (reviewer **9BHQ**); that “the division between discrete and continuous latent concepts is conceptually clean and helps generalize across types of reasoning,” and that we “situate [our] findings well within recent discussions of task vectors, linear representation hypothesis, and in-context reasoning circuits” (reviewer **e7J6**).

At the same time, we acknowledge the main **critical points** raised across the reviews. We find that they can be roughly divided into two categories: one involving clarifications on our results and writing, and one involving limitations of our analytical setup due to our limited computing resources. We discuss them at a high level here, and respond in greater detail in our individual responses.

**Clarifications and writing improvement**. Reviewers asked for a clearer connection between the “concept maps” introduced in Section 1.1 and the technical procedures and results in Sections 2 and 3, as well as stability/statistical analyses beyond the ones presented in the main text.

In the revised manuscript, we more explicitly link the concept maps to our experimental procedures and findings, with the main changes highlighted in blue. We also emphasize that Appendices A and B contain the full details of our experimental results. Figures 12 - 26, and 28 - 30 present results of our activation patching sweeps over all source and target types on our multi-hop ICL datasets, providing _exhaustive causal evidence_ for the existence and utilization of hidden concept maps in the LLM. Figure 27 provides more systematic _correlational evidence_ for concept disentanglement in the model. Figures 31, 32 and 34 expand on the numerical-concepts experiments. We made more explicit reference to these results in the main text, again highlighted in blue.

**Limitations of analytical setup**. Reviewers raised concerns about the synthetic/idealized nature of our tasks and their generalization to more naturalistic reasoning tasks in ICL, and that we focused on a single LLM family (Gemma-2) plus a small transformer rather than a broader range of model families. We acknowledge this limitation. However, we wish to kindly note that, we prioritized our limited resources on what we believe to be the most important first steps for rigorously understanding the _mechanisms_ by which transformers can _disentangle and manipulate latent concept representations during in-context learning (ICL)_.

To elaborate, the controlled settings of our 2-hop reasoning and continuous-parameterization ICL tasks allow us to keep the ground truths _unambiguous_, perform precise _causal interventions_, and analyze the _geometry_ of representations. This clean setup enabled us to obtain, on the 2-hop ICL dataset, _**consistent localization** of concept invocation and composition mechanisms_ in the LLM, a _**scaling trend** of disentanglement with respect to model size_ in the LLM family; and, on the numerical-concepts datasets, clear evidence of the _order-preserving nature of the task representation manifolds_. We believe that achieving this level of clarity would have been substantially harder had we spread our resources across more LLM families and larger, less controlled datasets. The more controlled approach allows systematic experimentation and minimizes confounding variables, and we view it as a foundation for future work on more complex and naturalistic scenarios.

---

> ### Author Response · Authors · 2025-12-04
> **Preliminary evidence of causal transferability from ICL puzzles to more naturalistic settings**
>
> A relatively common question from the reviews is whether the latent concept embeddings localized in our synthetic ICL puzzles are specific to those tasks, or whether they reflect mechanisms that transfer to more naturalistic settings.
>
> While fully resolving this question is beyond the scope of our work and resources, we ran *small-scale additional experiments* that provide **preliminary but positive evidence that, the concept embeddings found in the ICL puzzle setting are *not* mere artifacts of the artificial setup: they exhibit precise causal influence on generation in more naturalistic settings**.
>
> The detailed setup and results are shown in **Appendix A.4**.
>
> **1. Experimental setup**
>
>
> We continue to work with Gemma-2-27B. We extract “Country” concept vectors solely from our synthetic ICL puzzles (as described in Section 2), using only the City→Capital source-target type. To test their transfer to naturalistic settings, we inject these vectors at attention layer 24 (as localized in the ICL setting) when the model is given open-ended, natural-sentence prompts.
>
>
> The natural prompts are constructed as follows:
> - A short *“hint” sentence* that implicitly describes a country, e.g.,
>  “From cherry blossoms to temples, everything here is unique.” (Japan)
>
>
> - A *“`Type`” sentence*, with `Type` $\in$ {`Universities`, `Celebrities`, `Tourist Spots`}, e.g.,
>  “I also know of some celebrities there, such as”
>
>
> An example prompt is: “From cherry blossoms to temples, everything here is unique. I also visited some great tourist spots there, such as”
>
> We focus on a small set of countries with a strong presence on the internet (e.g., US, UK, China) for the sake of tractability.
>
> **2. Results: causal influence in naturalistic prompts**
>
> Briefly speaking, we use an LLM judge (gpt-4o-mini) to evaluate whether an intervention causes the continuation to (i) switch from the source country implied by the context to the target country encoded by the concept vector, and (ii) remain specific to the chosen `Type`. The judge returns a relevance score from 1 (irrelevant to the target country/Type) to 5 (highly relevant). We then manually correct these scores. We observe the following.
>
>
> - Pre-intervention: Continuations naturally stay in the source country, with average target-country relevance  ~1.0.
>
>
> - Post-intervention: *For a substantial fraction of cases (roughly 60 to 80 %, depending on Type), the 27B model’s continuations are judged as describing the target country while remaining on the specified `Type`*.
>
>
> (See Figure 28 in the updated paper.)
>
>
> Importantly, the generations remain coherent; the interventions do not simply inject country tokens, but produce fluent text that *integrates the target concept within the given `Type`*. We present two examples here; please see Table 1 in the paper for a list of examples.
>
> *Example 1 (Tourist spots)*: China $\to$ South Korea
> - Prompt: I loved everything about my trip - the high-speed rail, the tea culture, and especially the lanterns. I also visited some great tourist spots there, such as
> - Natural continuation: the 99 Dragons Wall, the Xiamen University campus, and the Huandao Road.
> - Intervened continuation: the Jeju Olle Trail, Seongsan Sunrise Peak, and Udo Island.
>
> *Example 2 (Celebrities)*: China $\to$ UK
> - Prompt: From lanterns to tea culture, everything here is unique. I also know of some celebrities there, such as
> - Natural continuation: Zhang Yimou and Gong Li. I love the food there, like hot pot and dumplings.
> - Intervened continuation: George Harrison, who visited this place. It was the time when The Beatles were rising to stardom
>
> These results suggest that the **_concept embeddings discovered in the ICL puzzle setting_** can also behave as meaningful control directions for country-level concept in more naturalistic text, suggesting that they are, at least in our setting, **_disentangled from surface linguistic patterns_**.
>
> 3. **Scaling (model size and disentanglement)**
>
> We also repeated the same intervention experiments on **Gemma-2-2B** (intervening at layer 15, which showed weaker but non-trivial causal scores in our ICL experiments before; see Figure 26). As shown in Figure 29, the fraction of post-intervention continuations receiving score 5 from the judge is roughly 1.5 to 2 times lower for the 2B model than for the 27B model. **While still preliminary, this is consistent with our ICL-setting findings and supports the interpretation that concept disentanglement improve with model scale.**
>
> Finally, we believe that understanding concept disentanglement is an important open problem in mechanistic interpretability: works which aim to understand how causally generalizable the mechanisms are across strict/synthetic and naturalistic context types are particularly scarce. We aim to expand upon the results in this update in the final version of the paper and open-source our code and datasets upon acceptance of our work to facilitate deeper studies in this direction.

---

### Meta-Review · Area_Chair_3TGT · 2026-01-07

**Summary:**

the paper provides a systematic and well-executed analysis of how LLMs represent and manipulate latent concepts. The primary strengths noted across the board include the creative use of causal mediation analysis (CMA) and activation patching to move beyond mere correlation, as well as the novel demonstration of low-dimensional manifold structures in representation space.

Main concern: Reviewers questioned whether these "toy" findings would generalize to naturalistic reasoning or across diverse model architectures beyond the Gemma-2 family.

**Reviewer Concerns:**

Addressed:

- Stability and Statistical Significance (qgy5)
- Clarified the link between the causal model in the introduction and the implemented CMA pipeline
- "toy" nature of the setup. (partially)

Outstanding:
- Theoretical Grounding (4uha): why transformers naturally converge on linear geometric task vectors
- Complexity of Real-World Concepts (4uha)

**Reviewer Scores:**

Remain unchanged.

---

### Decision · Program_Chairs · 2026-01-26

Accept (Poster)